# Crown-hydroxylamines are pH-dependent chelating *N,O*-ligands with a potential for aerobic oxidation catalysis

Vladislav K. Lesnikov [1], Ivan S. Golovanov [1], Yulia V. Nelyubina[2,3], Svetlana A. Aksenova [2,3] & Alexey Yu. Sukhorukov [1] ✉

Despite the rich coordination chemistry, hydroxylamines are rarely used as ligands for transition metal coordination compounds. This is partially because of the instability of these complexes that undergo decomposition, disproportionation and oxidation processes involving the hydroxylamine motif. Here, we design macrocyclic poly-*N*-hydroxylamines (crown-hydroxylamines) that form complexes containing a d-metal ion (Cu(II), Ni(II), Mn(II), and Zn(II)) coordinated by multiple (up to six) hydroxylamine fragments. The stability of these complexes is likely to be due to a macrocycle effect and strong intramolecular H-bonding interactions between the N−OH groups. Crown-hydroxylamine complexes exhibit interesting pH-dependent behavior where the efficiency of metal binding increases upon deprotonation of the hydroxylamine groups. Copper complexes exhibit catalytic activity in aerobic oxidation reactions under ambient conditions, whereas the corresponding complexes with macrocyclic polyamines show poor or no activity. Our results show that crown-hydroxylamines display anomalous structural features and chemical behavior with respect to both organic hydroxylamines and polyaza-crowns.

Macrocyclic polyamines (polyaza-crowns, Fig. 1a) have tremendously impacted many areas, including coordination chemistry, catalysis, nanotechnology, medicinal, and materials chemistry[1–6]. The ability of these ligands to strongly bind cations of transition metals, lanthanides and actinides, as well as small molecules, provides vast opportunities for applications in the fields of catalysis, photo- and electrocatalysis, separation, recognition, sensing, medical diagnostics, and therapy[7–10]. One of the most known applications is the development of contrast agents (e.g., Gadoteric acid[11]) and targeted radiopharmaceuticals based on DOTA-derived complexes[12,13]. Macrocyclic amines, tacn, cyclam and cyclen in particular, proved to be convenient platforms for the design of sensors for cations[14], anions and physiologically relevant small molecules[14,15]. Transition metal complexes of macrocyclic amines

mimic the structure and functions of hemoproteins and other metalloenzymes[16–21]. Iron complexes with cyclam-derived ligands were shown to catalyze hydrogen and oxygen atom transfer reactions via oxoiron(IV) species (such as $[Fe^{IV}(O)(tmc)(NCMe)]^{2+}$) formed via dioxygen activation[22–24].

The properties of polyaza-crowns can be efficiently tuned by the macrocycle size and, more importantly, by substitution at the nitrogen atoms[25,26]. Although numerous *N*-substituted macrocyclic amines have been prepared and extensively studied, *N*-hydroxy derivatives remain unexplored. In fact, only two reports deal with partially *N*-hydroxylated derivatives of macrocyclic amines containing a maximum of two N−OH groups in the macrocycle[27,28]. To the best of our knowledge, no examples of fully *N*-hydroxylated

[1]N. D. Zelinsky Institute of Organic Chemistry, Russian Academy of Sciences, 119991 Leninsky prospect, 47, Moscow, Russian Federation. [2]A. N. Nesmeyanov Institute of Organoelement Compounds, Russian Academy of Sciences, 119991 Vavilova str. 28, Moscow, Russian Federation. [3]Moscow Institute of Physics and Technology (National Research University), 141700 Institutskiy per. 9, Dolgoprudny, Moscow Region, Russian Federation. ✉e-mail: sukhorukov@ioc.ac.ru

**a.** Macrocyclic polyamines (polyaza-crowns):

tacn          cyclam          cyclen          [20]-ane[NH]₅

Gadoteric acid
(contrast agent)

[Fe$^{IV}$(O)(tmc)(NCCH₃)]$^{2+}$
(model of nonheme Fe enzymes)

**b.** Hydroxylamines as ligands:

- switch between N,O-coordination modes
- possibility of $\eta^2$-N,O-coordination
- strong H-bond donor/acceptor
- redox-active motif
- stabilization of d-metal higher oxidation states

tfoH₃

O³-ligand

[M$^{IV}$(tfo)L₃]$^+$
aqueous stable
Fe(IV), Mn(IV) and Ni(IV)

**c.** Macrocyclic polyhydroxylamines (hitherto unknown, focus of this work):

- synthesis
- structure
- coordination with d-metals
- pH-dependence
- aerobic oxidation catalysis

**Fig. 1 | The background for this study. a** Structures of common macrocyclic polyamine ligands (polyaza-crowns) and examples of their metal complexes. **b** Hydroxylamines as ligands and examples of their metal complexes. **c** Macrocyclic polyhydroxylamines (focus of this work).

macrocyclic polyamines or their derivatives have been reported in the literature so far.

Fundamentally different properties can be anticipated upon incorporation of hydroxyl groups at the nitrogen atoms of macrocyclic polyamines. Indeed, hydroxylamines exhibit a versatile coordination chemistry that differs substantially from that of amines, not only in terms of the metal binding selectivity, but also in the coordination modes (Fig. 1b)[29,30]. Hydroxylamines can coordinate metal ions with either nitrogen or oxygen atoms, or both ($\eta^2$-coordination)[29,31]. Due to an enhanced acidity of the N–OH fragment, the coordination mode can change from one protonation state to another[32]. Hydroxylamines form strong hydrogen bonds[33] that contribute to the stability of metal complexes[34]. An intriguing feature of hydroxylamine ligands is their redox non-innocence, which affects the reactivity of the corresponding metal complexes[30]. Nitroxyl radicals (TEMPO, ABNO) that result from oxidation of cyclic hydroxylamines are extensively used in combination with d-metals as catalysts for selective oxidation of alcohols and other organic compounds under aerobic conditions[35,36]. Well-defined metal-nitroxide and metal-hydroxylamine complexes involved in these catalytic cycles were isolated and structurally characterized by Captain et al.[37], Lancaster et al.[38], Hayton et al.[39], Ottenwaelder et al.[30], and others[36]. In these catalysts, the N–O bond can intramolecularly assist the oxidation of an organic substrate via nitroxyl radical species[30]. Moreover, deprotonated hydroxylamine ligands can stabilize higher oxidation states of d-metals. For example, our group recently demonstrated that the simplest cyclic trishydroxylamine (tfoH₃) coordinated d-metals with oxygen atoms to give well-defined Fe(IV), Mn(IV) and Ni(IV) complexes stable in aqueous solutions[40]. Nevertheless, structurally characterized complexes in which the metal ion is coordinated by more than three hydroxylamine groups (of non-oxime nature) are rare[32,41]. Those few reported in the literature easily undergo oxidation and disproportionation under ambient conditions. The design of stable poly-hydroxylamine

complexes would provide further impetus to studies on the reactivity and potential applications of metal-hydroxylamine species in catalysis and metal ion sensing.

In this work, we wish to report the synthesis of a series of fully *N*-hydroxylated macrocyclic polyamines (crown-hydroxylamines) containing from three to five hydroxylamine units (Fig. 1c). We have shown that crown-hydroxylamines exhibit anomalous structural features and behavior toward both organic hydroxylamines and polyaza-crowns. Crown-hydroxylamines form air-stable complexes, in which the transition metal ion (Cu(II), Ni(II), Mn(II) or Zn(II)) is coordinated by multiple (up to six) hydroxylamine groups. These complexes exhibit interesting pH-dependent behavior where the efficiency of metal binding increases upon deprotonation of hydroxylamine groups. Copper complexes of crown-hydroxylamines show catalytic activity in dioxygen activation, whereas the corresponding complexes with macrocyclic polyamines show poor or no activity in these reactions.

## Results

### Synthesis of macrocyclic *N*-hydroxylamines and their derivatives

At the outset, direct *N*-hydroxylation of parent macrocyclic amines was thought to be a logical approach towards macrocyclic *N*-hydroxylamines. Numerous methods for *N*-hydroxylation of secondary amines are available in the literature[42–44], most of which employ peroxide-based reagents (H₂O₂, mCPBA, Oxone, ᵗBuOOH/silica gel). Unfortunately, our attempts to perform *N*-hydroxylation of a model macrocyclic tetramine (cyclam) using these procedures failed and resulted in indecipherable mixtures of overoxidation products. Eventually, we switched to another strategy that relied on the *N*-acyloxylation of amines with diacyl peroxides followed by hydrolysis of the acyl groups. This method initially discovered by Gambarjan[45] and recently modified by Yamamoto[46] was never tested for polyamines. We

were happy to find that the reaction of cyclam with excess benzoyl peroxide (8 equiv.) and Cs₂CO₃ (12 equiv.) delivered the desired tetra-*N*-benzoyloxylated derivative cyclam(OBz)₄ in an acceptable yield (54%, Fig. 2a, entry 1). A side product Bz-cyclam(OBz)₃ was also formed, which contained one *N*-benzoylated and three *N*-benzoyloxylated units. The formation of this product could be largely suppressed by performing the reaction in the presence of water (entries 2, 3 in Fig. 2a). The use of K₂CO₃ instead of Cs₂CO₃ resulted in a reduction in the yield of the target cyclam(OBz)₄ demonstrating a pronounced cesium effect (entry 4). However, the amount of expensive Cs₂CO₃ could be reduced to 1 equiv. by substituting most of it with K₂CO₃ (11 equiv.). Under these conditions (entry 5), cyclam(OBz)₄ was obtained in 75% yield with only traces of side product Bz-cyclam(OBz)₃. A

decrease in the amount of benzoyl peroxide led to a substantial drop in the reaction efficiency (entry 6, Fig. 2a).

Next, we tested a series of 9−20 membered macrocyclic amines containing three to five nitrogens in the *N*-benzoyloxylation reaction (method A or method B, Fig. 2b). To our delight, this was a general process that gave the desired *N*-benzoyloxylated macrocyclic polyamines. For some macrocyclic amines (especially, tacn), better results were obtained when excess Cs₂CO₃ was used (method A) in comparison to method B. Although the yields were moderate in many cases, it should be emphasized that the process involves multiple independent addition reactions, and the partial yields per one NH-group are rather high. The only unsuccessful example was cyclen, where the desired tetra-benzoylated product cyclen(OBz)₄ was not obtained. This could

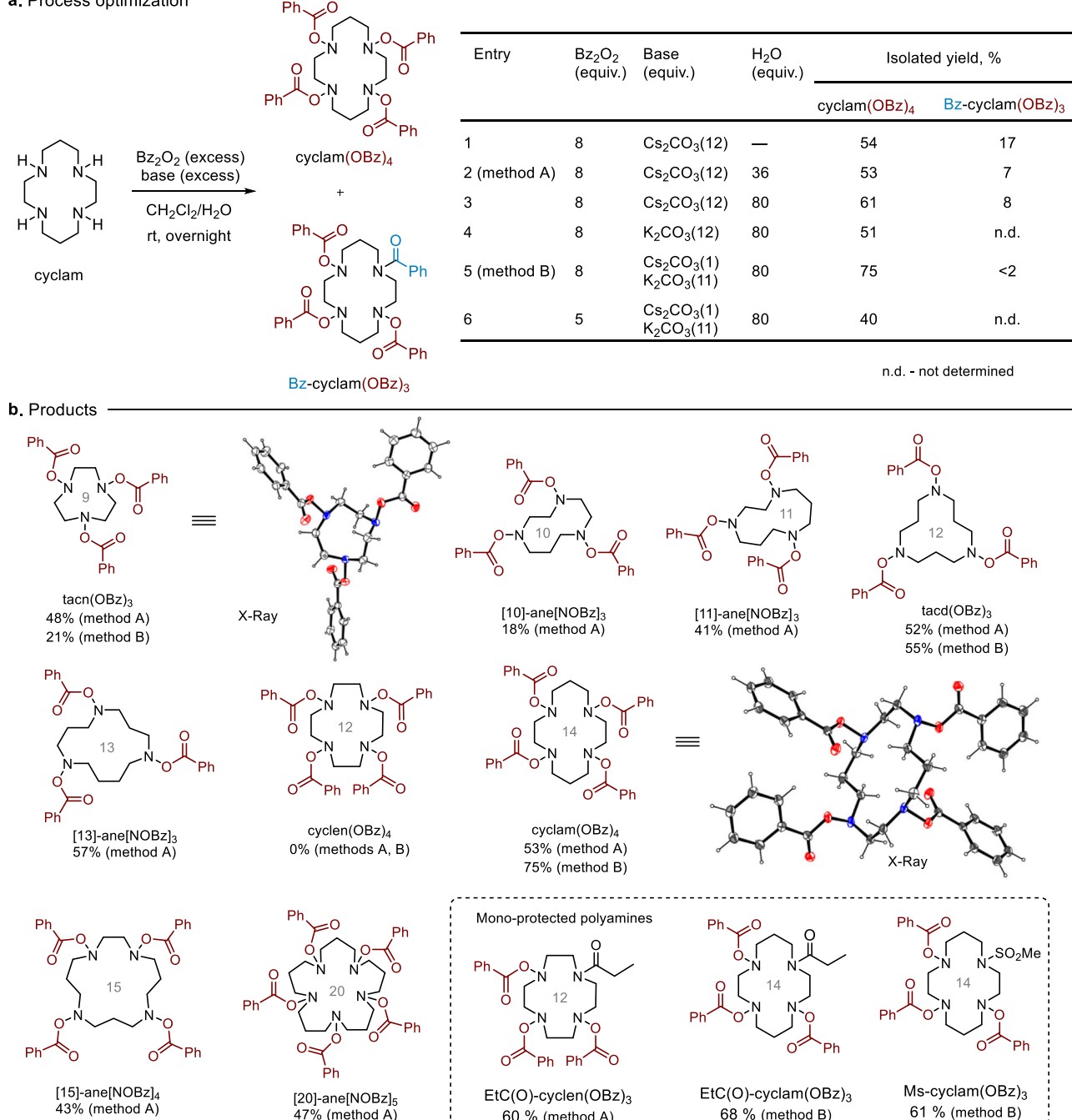

**Fig. 2 | Synthesis of *O*-benzoyl-protected macrocyclic hydroxylamines. a** Process optimization. **b** Structures and yields of the products.

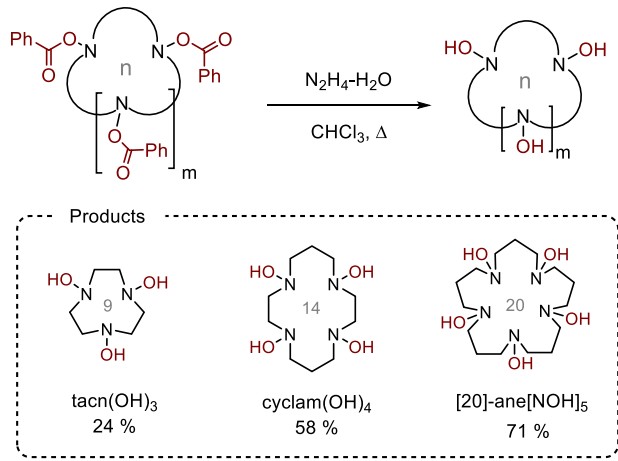

**Fig. 3 | Synthesis of macrocyclic poly-*N*-hydroxylamines by unmasking the benzoyl group in Bz-protected macrocyclic hydroxylamines.** Reaction conditions for tacn(OH)$_3$: tacn(OBz)$_3$ (0.22 mmol), hydrazine hydrate (3.3 mmol), CHCl$_3$ (3 ml), rt, overnight (argon atmosphere). Reaction conditions for cyclam(OH)$_4$: cyclam(OBz)$_4$ (0.98 mmol), hydrazine hydrate (20 mmol), CHCl$_3$ (17 ml), reflux, 4 h. Reaction conditions for [20]-ane[NOH]$_5$: [20]-ane[NOBz]$_5$ (0.07 mmol), hydrazine hydrate (1.8 mmol), CHCl$_3$ (3 ml), reflux, 4 h. The yields refer to the isolated products.

be attributed to some conformational effects leading to a reduced reactivity of one of the nitrogen atoms. Indeed, the mono-protected *N*-propyonyl-cyclen derivative (EtC(O)-cyclen) produced the corresponding tris-benzoylated product EtC(O)-cyclen(OBz)$_3$ (Fig. 2b). *N*-Benzoyloxylation of mono-protected cyclam derivatives (*N*-propyonyl and *N*-mesyl derivatives) was performed in a similar fashion. The successful synthesis of products with one masked secondary amino group shows that macrocyclic *N*-hydroxylamines bearing a functional tag can be prepared using the developed strategy.

At the next stage, unmasking of the *N*-benzoyloxy-groups was performed. After testing several common reagents for the removal of the benzoyl group (LiOH, K$_2$CO$_3$, NH$_3$, N$_2$H$_4$, see Supplementary Table 1), we found that treatment of cyclam(OBz)$_4$ with excess hydrazine hydrate in refluxing chloroform afforded the desired tetra-hydroxylamine derivative cyclam(OH)$_4$ in 58% yield (Fig. 3a). This crystalline compound (mp 210−213 °C) could be prepared on at least 150 mg scale and stored at 0−5 °C for at least 6 month without noticeable decomposition. It is soluble in water and DMSO, slightly soluble in methanol, and insoluble in chloroform and in non-polar organic solvents.

The 20-membered pentahydroxylamine [20]-ane[NOH]$_5$ was prepared by the same procedure in an even higher yield (71%) as a white solid material that precipitated from the reaction mixture. However, 9-membered cyclic tris-hydroxylamine tacn(OH)$_3$ was more soluble in organic solvents, and its separation from the benzhydrazide by-product was problematic. In addition, tacn(OH)$_3$ underwent oxidative degradation upon column chromatography. A similar behavior was observed for tris-hydroxylamine [12]-ane[NOH]$_3$ derived from the 12-membered ligand tacd. Despite these instability issues, the unprotected macrocyclic tris-hydroxylamines [n]-ane[NOH]$_3$ could be generated in situ from the corresponding stable benzoates as demonstrated below.

## Structure of macrocyclic hydroxylamines

The structures of benzoates tacn(OBz)$_3$ and cyclam(OBz)$_4$ were secured by X-ray diffraction analysis (Fig. 2). In the crystal state of tris-benzoyloxyamine tacn(OBz)$_3$, the macrocycle conformation is similar to that of the parent tacn. The 14-membered macrocycle in cyclam(OBz)$_4$ adopts an exodentate conformation with *trans*-IV

configuration, which is not characteristic of neutral cyclam derivatives (the parent cyclam has an endodentate *trans*-III conformation)[47]. All the *N*-benzoyloxy groups accommodate pseudo-axial positions.

Macrocyclic hydroxylamines did not form suitable crystals for X-ray analysis. Nevertheless, we succeeded in preparing single crystals of mono- and diprotonated salts of cyclam(OH)$_4$ (hydrochloride and dihydrobromide). The structure of the [cyclamH$_2$(OH)$_4$]$^{2+}$ dication (Fig. 4a) resembled the diprotonated cyclam with the endodentate conformation of the macrocycle and two intramolecular hydrogen bonds (N1−H...N1 2.833(8) Å, 137.3°). All the hydroxyl groups are located in pseudo-axial positions and form hydrogen bonds with the bromide anions (O1−H...Br1 3.182 Å, angle 156.6°).

The structure of cyclam(OH)$_4$ mono-hydrochloride is more remarkable (Fig. 4b). The macrocycle has a *trans*-IV conformation with one of the non-protonated hydroxylamine units existing in an unusual tautomeric *N*-oxide form. The tautomer of unsubstituted hydroxylamine (ammonia oxide, H$_3$N$^+$−O$^-$) was detected in the crystal state[48,49], while its presence in solution has been argued[50]. Tautomers of organic hydroxylamines were not previously observed, albeit few examples of their metal complexes were reported[37]. In the [cyclamH(OH)$_4$]$^+$ cation, the semipolar N$^+$−O$^-$ bond distance (1.403(2) Å) is noticeably shorter than that of the N−O bond in the non-tautomerized hydroxylamine unit (1.452(2) Å) and is close to the N$^+$−OH bond (1.412(2) Å) in the protonated hydroxylamine unit. The N$^+$−O$^-$ bond distance is also close to that observed previously in ammonia oxide (1.4170(15) Å) by Kirby and Nome et al.[48]. The N$^+$−H unit of the tautomeric hydroxylamine is involved in hydrogen bonding with the nitrogen atom of the neighboring NOH fragment (N1−H1...N4 2.849(2) Å, 109(1)°). The *N*-oxide oxygen also forms a hydrogen bond with a lattice methanol molecule (Fig. 4b). We believe that these hydrogen bonding interactions (especially, the intramolecular N1−H1...N4 bond) play a pivotal role in stabilizing the *N*-oxide tautomeric form. We may speculate that the formation of tautomeric hydroxylamine forms may be a common feature of crown-hydroxylamines since their macrocyclic structure favors intramolecular interactions stabilizing the *N*-oxide form. Hence, macrocyclic *N*-hydroxylamines can serve as models to explore the reactivity of ammonia oxide.

In a solution (water), macrocyclic hydroxylamines tacn(OH)$_3$, cyclam(OH)$_4$ and [20]-ane[NOH]$_5$ show broadened signals in the $^1$H and $^{13}$C NMR spectra. This can be attributed to slow dynamic processes associated with either macrocycle conformation changes, inversion of nitrogen atoms, or the aforementioned tautomerization of hydroxylamine groups. Since the parent macrocyclic amines give sharp signals, we believe that the ring flipping is fast at ambient temperature. Indeed, in the $^1$H NMR spectra of cyclam only two sets of signals are observed suggesting that the axial and equatorial hydrogens are indistinguishable (Fig. 5a). In cyclam(OH)$_4$, the hydrogens in equivalent CH$_2$ units are separated, indicating that the conformation of the macrocycle is more fixed, apparently, due to a strong intramolecular hydrogen bonding (Fig. 5a, b).

The basicity of free cyclam(OH)$_4$ was studied by potentiometric pH titration (*I* = 0.01 M, KCl). Four equilibrium constants were derived from the pH profile of the ligand (5.5, 4.0, 3.2 and 2.6), which correspond to the protonation of the nitrogen atoms. Indeed, the first constant is close to the p$K_a$(BH$^+$) value of *N*,*N*-diethylhydroxylamine (5.6) in H$_2$O[51]. Expectedly, the observed constants are lower than those reported[52] for cyclam (11.6, 9.6, 3.6, 2.6) due to the reduced basicity of hydroxylamines. For this reason, tris- and tetra-protonated salts of cyclam(OH)$_4$ were not formed in the presence of excess acid, although they are known for cyclam.

## d-Metal complexes of tacn(OH)$_3$

We started the exploration of the coordination chemistry of macrocyclic hydroxylamines with *N*,*N*,*N*-trihydroxy-1,4,7-triazacyclononane

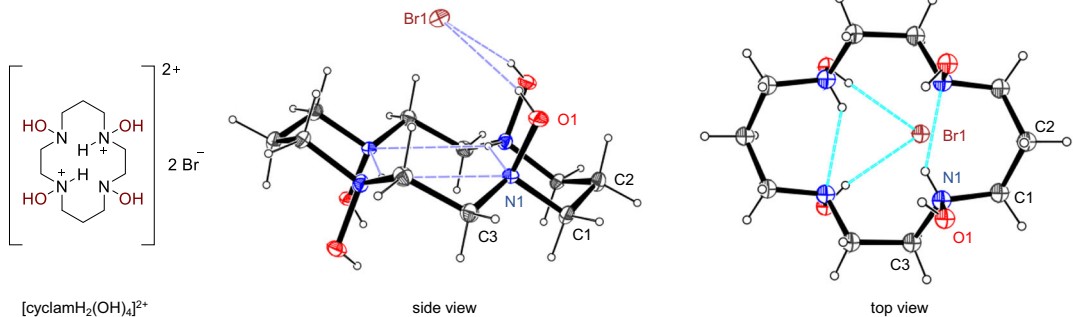

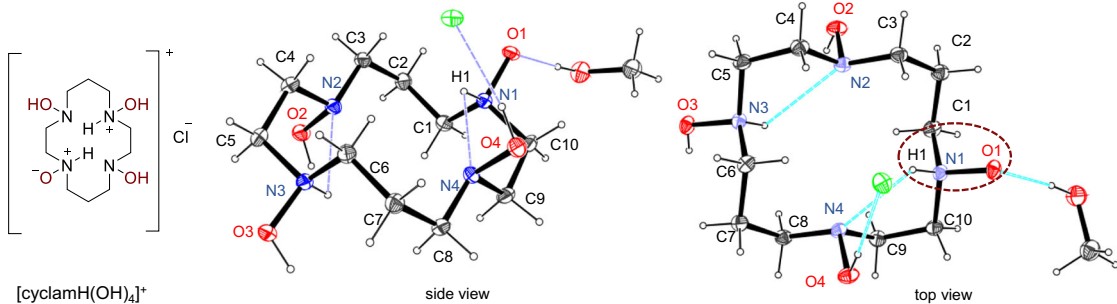

**Fig. 4 | Crystal structures of cyclam(OH)₄ salts. a** Two ORTEP views of [cyclamH₂(OH)₄]²⁺ 2Br⁻. Selected bond lengths (Å) and angles (°): N1−O1 1.445(5), N1−C1 1.484(7), N1−C3 1.485 (7), C1−N1−O1 106.0 (4). **b** Two ORTEP views of [cyclamH(OH)₄]⁺ Cl⁻·MeOH. Selected bond lengths (Å) and angles (°): N1−O1

1.403(2), N2−O2 1.457(2), N3−O3 1.412(2), N4−O4 1.452(2), N1−C1 1.496(2), N1−C10 1.501(3), N4−C9 1.459(2), O1−N1−C10 108.2(1), O4−N4−C9 106.3(1). Displacement ellipsoids are drawn at 50% probability level.

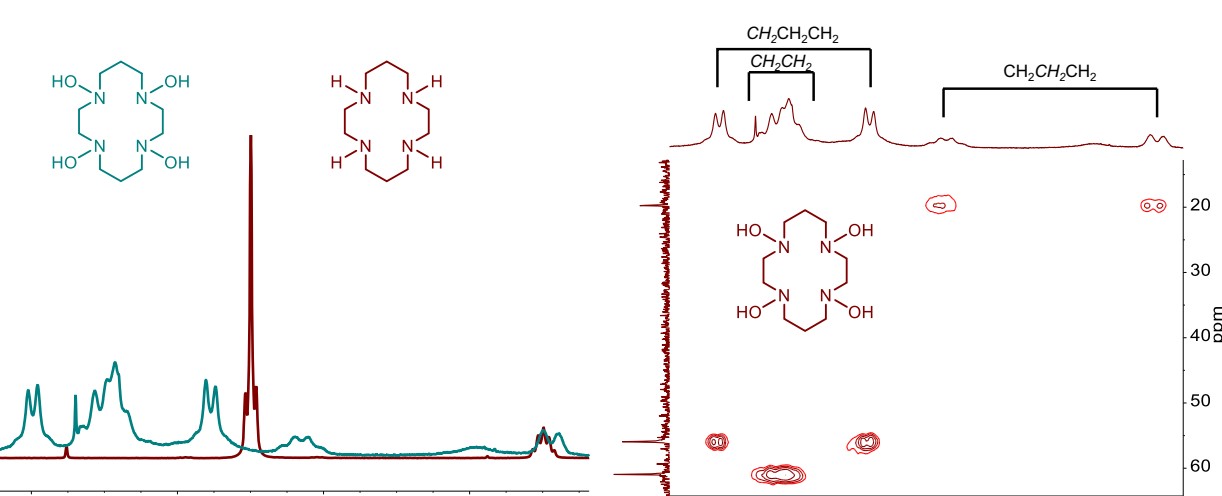

**Fig. 5 | Comparison of NMR spectra of cyclam and cyclam(OH)₄. a** Superimposition of ¹H NMR spectra of cyclam and cyclam(OH)₄ taken in D₂O at rt. **b** A fragment of the ¹H-¹³C HSQC spectrum of cyclam(OH)₄.

tacn(OH)₃ (Fig. 6). Since tacn(OH)₃ was difficult to isolate in pure state, the corresponding benzoate derivative tacn(OBz)₃ was used in these studies. We were pleased to find that the reaction of tacn(OBz)₃ with NiCl₂•6H₂O in methanol gave a stable complex [Ni₂(μ-Cl)(μ²-O₂CPh)(tacn(OH)₃)₂Cl₂] that contained fully deprotected tacn(OH)₃ as the ligand (Fig. 6a, b). This shows that the benzoate moiety can be cleaved under mild conditions upon coordination of *N*-atoms with the metal.

The role of the metal cation is essential in this case, since tacn(OBz)₃ is stable in methanol under ambient conditions. Apparently, the binding of a metal ion to the protected ligand makes the benzoate groups more electrophilic and liable to methanolysis/hydrolysis (Fig. 6a).

The binuclear nickel-tacn(OH)₃ complex comprises two octahedral Ni(II) cations bridged via one carboxylate anion and one chloride anion (Fig. 6c). The nickel cations are additionally coordinated by

**Fig. 6 | Complexation of tacn(OH)$_3$ with transition metals. a** In situ deprotection of tacn(OBz)$_3$ enabled by coordination with metal ions. **b** Synthesis of nickel(II) and zinc(II) complexes with tacn(OH)$_3$. **c** Crystal structure of [Ni$_2$(μ-Cl)(μ$^2$-O$_2$CPh)(tacn(OH)$_3$)$_2$Cl$_2$]. Selected bond lengths (Å) and angles (°): Ni1−N1, 2.113(3), Ni1−N2 2.084(2), Ni1−N3, 2.095(2), Ni1−O7 2.006(3), Ni1−Cl1 2.4226(9), Ni1−Cl2 2.4043(9), N1−O1 1.451(4), N3−O3 1.438(3), O1−N1−C1 106.8(2), Ni1−Cl2−Ni2 109.10(3).

**d** Crystal structure of [Zn(tacn(OH)$_3$)$_2$](NO$_3$)$_2$ (ORTEP and space-filling representation). Selected bond lengths (Å) and angles (°): Zn1−N1 2.2442(8), Zn1−N2 2.1564(9), Zn1−N3 2.1852(8), N1−O1 1.439(1), N2−O2 1.4477(11), N3−O3 1.4379(10), N1−Zn1−N2 78.38(3), N2−Zn1−N3' 105.27(3), N2−Zn1−N2' 170.16(4). Displacement ellipsoids are drawn at the 50% probability level, hydrogens at carbon atoms of tacn(OH)$_3$ are omitted for clarity.

chloride anions and three nitrogen atoms of tacn(OH)$_3$. The macrocycle conformation, as well as the Ni−N (2.084(2)−2.113(3) Å) and Ni···Ni (3.9258(6) Å) distances, are similar to those of the parent [Ni(tacn)$_2$]Cl$_2$ and [Ni$_2$(μ-Cl)$_2$(tacn)$_2$Cl$_2$] complexes. A remarkable feature of the nickel-tacn(OH)$_3$ complex is the formation of multiple intramolecular O−H...Cl hydrogen bonds involving hydroxylamine units and chlorine atoms (both bridging Cl2 and non-bridging Cl1 and Cl3; the distances are: O−H...Cl 3.141(3)−3.186(5) Å and 3.064(3)−3.167(3) Å, respectively).

In the reaction of benzoate tacn(OBz)$_3$ with zinc nitrate, a stable 2:1 complex [Zn(tacn(OH)$_3$)$_2$](NO$_3$)$_2$ was obtained (Fig. 6b). X-ray analysis revealed a sandwich-like dicationic structure [Zn(tacn(OH)$_3$)$_2$]$^{2+}$ (Fig. 6d). To the best of our knowledge, this is the only structurally characterized complex in which the metal ion is coordinated by six hydroxylamine groups. Again, a feature of the complex was that intramolecular hydrogen bonds formed between the hydroxyl groups of both macrocyclic ligands. Two pairs of NOH groups are involved in these interactions (O2(2')−H...O1'(1) 2.6915(10) Å, 163.29(5)°). The third pair of hydroxyls O3−H and O3'−H pointed in opposite directions make intermolecular H-bonds with the

nitrate anions. However, a relatively close O...O contact is observed between these hydroxyl groups (O3...O3' 2.9031(14) Å). Through these non-covalent interactions, two tacn(OH)$_3$ ligands form a supramolecular cage in which the metal ion is encapsulated similarly to clathrochelates. The Zn−N2 and Zn−N3 distances in [Zn(tacn(OH)$_3$)$_2$]$^{2+}$ (2.1564(9)−2.1852(8) Å) are similar to those reported for [Zn(tacn)$_2$]$^{2+}$ (2.166−2.182 Å)[53], while the Zn−N1 bond (involved in the H-bond with the nitrate anion) is somewhat elongated (2.2442(8) Å).

Both nickel and zinc complexes of tacn(OH)$_3$ are air-stable, demonstrating the stabilization of the tris-hydroxylamine ligand to oxidation upon coordination with a metal ion.

## d-Metal complexes of cyclam(OH)$_4$

The fully N-hydroxylated 1,4,8,11-tetraazacyclotetradecane (cyclam(OH)$_4$) readily formed complexes with copper(II), nickel(II), manganese(II) and zinc(II) salts in an aqueous or methanolic medium. The corresponding crystalline M(cyclam(OH)$_4$)X$_2$ complexes with M = Cu (X = Cl), Mn (X = Cl, Br), Ni (X = NO$_3$, ClO$_4$), and Zn (X = Cl) were obtained and characterized by X-ray diffraction analysis. The geometries of the Cu(II), Mn(II) and Zn(II) complexes are very similar (Fig. 7a,

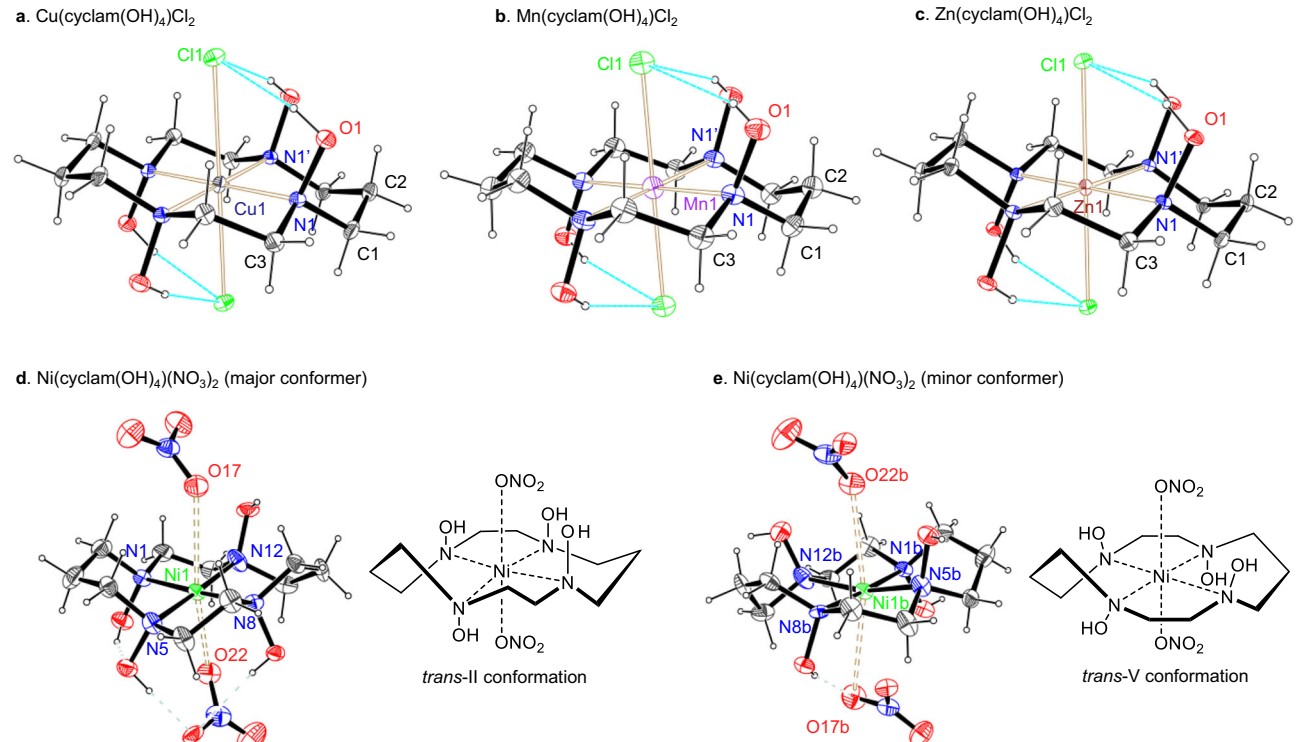

**a**. Cu(cyclam(OH)₄)Cl₂    **b**. Mn(cyclam(OH)₄)Cl₂    **c**. Zn(cyclam(OH)₄)Cl₂

**d**. Ni(cyclam(OH)₄)(NO₃)₂ (major conformer)    **e**. Ni(cyclam(OH)₄)(NO₃)₂ (minor conformer)

*trans*-II conformation    *trans*-V conformation

**Fig. 7 | Crystal structures of M(cyclam(OH)₄)Hal₂ complexes. a** Cu(cyclam(OH)₄)Cl₂. Selected bond lengths (Å) and angles (°): Cu1−N1 2.0511(14), N1−O1 1.4515(18), Cu1−Cl1, 2.7853(6), N1−Cu1−N1′ 93.98(8), N1−Cu1−Cl1 91.22(4). **b** Mn(cyclam(OH)₄)Cl₂. Selected bond length (Å) and angles (°): Mn1−N1 2.1943(18), N1−O1 1.445(2), Mn1−Cl1, 2.6136(8), N1−Mn1−N1′ 95.72(9), N1−Mn1−Cl1 92.88(5). **c** Zn(cyclam(OH)₄)Cl₂. Selected bond lengths (Å) and angles (°): Zn1−N1 2.1334(10), N1−O1 1.4489(13), Zn1−Cl1, 2.5921(4), N1−Zn1−N1′ 95.14(6), N1−Zn1−Cl1 88.38(3). **d** Ni(cyclam(OH)₄)

(NO₃)₂ (major conformer, 72%). Selected bond lengths (Å) and angles (°): Ni1−N1 1.961(6), Ni1−N5 1.928(5), Ni1−N8 1.933(6), Ni1−N12 1.942(7), Ni1−O17 2.945(6), Ni1−O22 2.598(7), O17−Ni1−O22 169.4(2), N1−Ni1−N5 93.1(2). **e** Ni(cyclam(OH)₄)(NO₃)₂ (minor conformer, 28%). Selected bond lengths (Å) and angles (°): Ni1b−N1b 1.928(17), Ni1b−N5b 1.970(13), Ni1b−N8b 1.995(15), Ni1b−N12b 2.074(17), Ni1b−O17b 2.73(2), Ni1b−O22b 2.917(15), O17b−Ni1b−O22b 172.1(6), N1b−Ni1b−N5b 91.6(5). Displacement ellipsoids are drawn at the 50% probability level.

b, c). The complexes are centrosymmetric; the metal ion is located exactly in the 4 N plane of the macrocycle, in the macrocycle center, and is coordinated by four nitrogens and two crystallographically equivalent halide ions. The M−Cl bonds are much longer than the M−N bonds and provide an axially elongated octahedral geometry around the metal center. As compared to the known [M(cyclam)]Hal₂ complexes[54,55], the M−N bonds in M(cyclam(OH)₄)Cl₂ units are elongated by 0.03−0.05 Å. The 14-membered ring of cyclam(OH)₄ is characterized by a *trans*-III geometry, with δ,λ-conformations of the five-membered and chair conformations of the six-membered chelate rings, which is the most stable and commonly found in cyclam complexes. The hydroxyl groups are located in axial positions (with respect to the macrocycle conformation) and the OH-bonds point at the halogen atoms, forming weak H-bond interactions (O1−H...Cl 2.9891(14) Å, 169.35(8)° for Cu, 3.033(2) Å, 158° for Mn, 2.9918(10), 159.42(6)° for Zn). Unlike cyclam complexes, in complexes of cyclam(OH)₄ all the hydrogen bonding contacts are intramolecular, while the molecular M(cyclam(OH)₄)Hal₂ units are organized into a 3D crystal network through numerous weak C−H...O contacts.

The geometry of the Ni(cyclam(OH)₄)(NO₃)₂ complex stands apart from the other M(cyclam(OH)₄)X₂ complexes (Fig. 7d, e). The Ni(II) center shows an axially elongated octahedron with two weakly bound nitrate anions (Ni...O 2.598(7)−2.945(6) Å vs 2.169 Å in the parent cyclam complex Ni(cyclam)(NO₃)₂[56]). Two conformers with highly uncommon *trans*-II (major, Fig. 7d) and *trans*-V geometries (minor, Fig. 7e) of the macrocyclic ring were present in the crystal[6]. Interestingly, the nickel ion and four nitrogen atoms are not located in one plane in the minor conformer.

Manganese complexes Mn(cyclam(OH)₄)Hal₂ (Hal = Cl, Br) are also of special note since the metal has the +2 oxidation state (Fig. 7b).

This situation is totally different from cyclam, which forms Mn(III) complexes spontaneously upon oxidation with air. Our attempts to prepare Mn(III) complexes of cyclam(OH)₄ directly from Mn(OAc)₃ gave complicated product mixtures which likely result from the oxidation of the ligand. Thus, the macrocyclic hydroxylamine ligand favors the Mn(II) oxidation state unlike the parent tetra-amine cyclam[57-60]. The Mn−N bonds (2.1943(18) Å for chloride and 2.172(3) Å for bromide) are considerably elongated in comparison to the known Mn(III) complexes of cyclam (2.029−2.042 Å[57]).

For copper, zinc and nickel M(cyclam(OH)₄)X₂ complexes, the logK values estimated by potentiometric pH titration (*I* = 0.01 M, KCl) were 12.6, 11.9 and 12.1 (at pH < 8). These values are lower than those for the corresponding M(cyclam)X₂ (24−27 for Cu, 15.5−17.0 for Zn and 20−22 for Ni[52]), demonstrating that the metal ion is bound more weakly by the four *N*-atoms of the hydroxylamine motifs under neutral conditions. Unexpectedly, in the competitive binding experiments with cyclam and cyclam(OH)₄, the nickel complex was formed preferentially with the latter (see Supplementary Fig. 28). This could indicate the presence of species other than [Ni(cyclam(OH)₄)]L₂, in which the nickel ion is bound more strongly than in the cyclam complex (*vide infra*).

## Cyclic voltammetry studies

To compare the redox chemistry of cyclam and cyclam(OH)₄, cyclic voltammetry (CV) measurements of free ligands and their complexes were performed in 0.1 M TBAPF₆ solution in DMSO. CV of free cyclam does not show any oxidation or reduction, while for cyclam(OH)₄ a strong anodic oxidation wave at $E_{ox}$ = 0.98 V and two irreversible reduction waves ($E_{red}$ = −0.66 and −1.16 V, vs. Fc/Fc⁺) were detected (Fig. 8a, b). The process is irreversible regardless of the scan rate indicating the $E_rC_i$ process. To identify the products of these

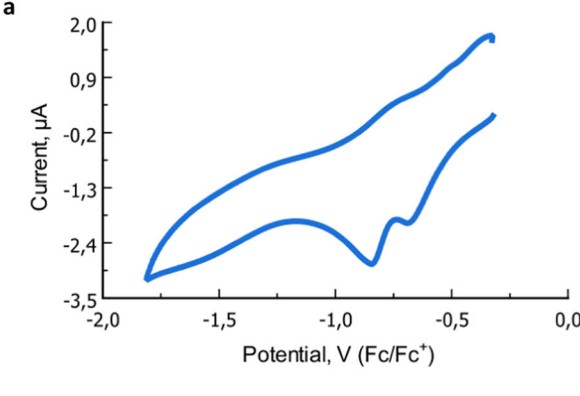

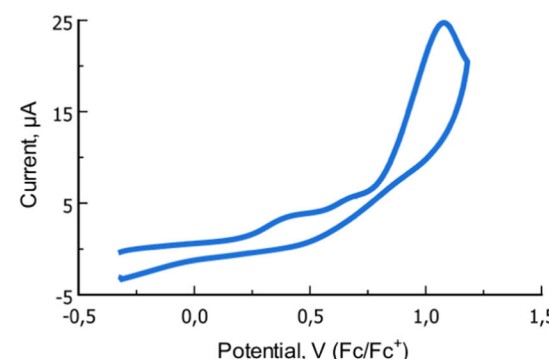

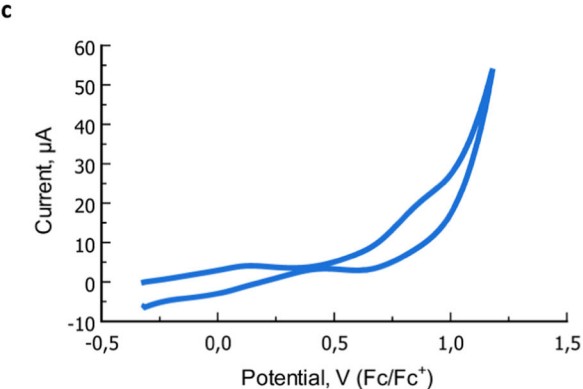

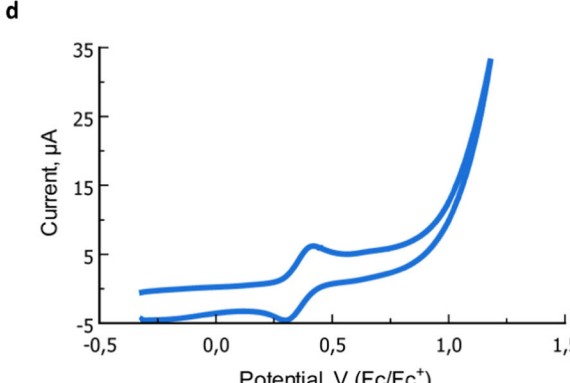

**Fig. 8 | Characteristic cyclic voltammograms.** (conditions: 50 mM TBAPF$_6$ in DMSO, with Pt disk working and counter electrodes, Pt wire ref. electrode, scan rate 100 mV/s, vs. Fc/Fc$^+$, E$_{1/2}$ (Fc/Fc$^+$) = 0.32 V). **a** Cyclic voltammogram of cyclam(OH)$_4$ for the reduction process. **b** Cyclic voltammogram of cyclam(OH)$_4$ for the oxidation process. **c** Cyclic voltammogram of Ni(cyclam(OH)$_4$)(ClO$_4$)$_2$ for the oxidation process. **d** Cyclic voltammogram of Ni(cyclam)(ClO$_4$)$_2$ for the oxidation process.

electron transfers, electrochemical oxidation was performed (1.5 V and 4 F/mol) followed by ESI-HRMS analysis. Ions corresponding to di-, tetra-, hexa- and octa-dehydrogenated ligand (L−2H, L−4H, L−6H, and L−8H) were identified. The same species were detected in a chemical oxidation of cyclam(OH)$_4$ by AgNO$_3$. NMR analysis showed signals around 7.5 ppm (protons) and 130 ppm (carbon) attributed to the double C,N-bonds. From these data, it can be suggested that cyclam(OH)$_4$ undergoes multiple irreversible oxidations of the hydroxylamine units to nitrones via nitroxyl radicals followed by abstraction of α-C−H hydrogens. However, attempts to isolate these oxidized products were unsuccessful most likely due to their instability.

CV of all M(cyclam(OH)$_4$)X$_2$ complexes displayed irreversible oxidation and reduction peaks regardless of the scan rate (see Fig. 8c and Supplementary Figs. 18, 21, 23, and 25). Albeit the nature of the observed redox processes is unclear (metal or ligand oxidation), the CV data suggest stabilization of the metal cation in oxidation state +2 in cyclam(OH)$_4$ complexes of Ni and Mn. The corresponding cyclam complexes undergo oxidation from M(II) to M(III) much easier (see Fig. 8d)[60]. Upon the addition of base (NaHCO$_3$), changes in CV are observed. In Ni(cyclam(OH)$_4$)(ClO$_4$)$_2$, the oxidation peak notably increases while its position remains the same (see Supplementary Fig. 22). This can be explained by the oxidation of a deprotonated complex whose concentration increases under basic conditions. In Cu(cyclam(OH)$_4$)Cl$_2$ the appearance and growth (upon multiple scanning) of a peak at $E_{ox}$ = −0.12 V is observed attributed to some unstable Cu(I) species generated during electrochemical processes (see Supplementary Fig. 19).

## pH-Dependent behavior of cyclam(OH)$_4$ complexes
Hydroxylamines are known to be weak acids, while coordination with a metal ion may increase the acidity. Upon deprotonation, hydroxylamine

becomes a much stronger σ-donating ligand, and its coordination properties may change dramatically. In this regard, the chemical properties of macrocyclic hydroxylamine complexes should be totally different from those of the parent macrocyclic amine complexes.

In an aqueous solution, the nickel(II)-cyclam(OH)$_4$ complex exhibited a remarkable behavior with an increase in pH. In a slightly acid medium, the complex is nearly colorless, while as the pH is increased, intense yellow color appears with a characteristic UV-Vis absorbance at 309 nm and 434 nm (Fig. 9a, b). The process is reversible and the yellow-colored species is converted into a colorless one upon addition of HCl. The formation of a nickel hydroxide precipitate was not observed even at pH 12. This situation substantially differs from nickel(II)-cyclam complexes, which are gradually decomposed under alkaline conditions with precipitation of nickel hydroxide[61]. NMR studies revealed that the colorless species are paramagnetic, which is consistent with the octahedral d[8] complex [Ni(cyclam(OH)$_4$)(H$_2$O)$_2$]$^{2+}$. In contrast, the yellow-colored complex shows a desymmetrized diamagnetic $^1$H and $^{13}$C NMR spectra. Electrospray high-resolution mass-spectrometry (ESI-HRMS) analysis showed the presence of the [Ni(cyclam(O$^-$)(OH)$_3$)]$^+$ ion as the major cationic species. Under neutral conditions, both paramagnetic and diamagnetic complexes are present in the solution as follows from the $^1$H NMR and UV-Vis data.

After numerous attempts to crystallize the deprotonated species, single crystals of a mixed neutral−deprotonated (1:1) complex were isolated and characterized by X-ray diffraction analysis (Fig. 9c). The cation [Ni(cyclam(O$^-$)(OH)$_3$)]$^+$ with a deprotonated ligand is characterized by a square-planar geometry around nickel, which is coordinated by four nitrogen atoms (Fig. 9c). The oxygen atom of the deprotonated hydroxylamine forms a strong hydrogen bond (O…O 2.519 Å) with a neighboring hydroxyl group, making a chelate structure somewhat similar to nickel(II) dioximate complexes[62]. The formation

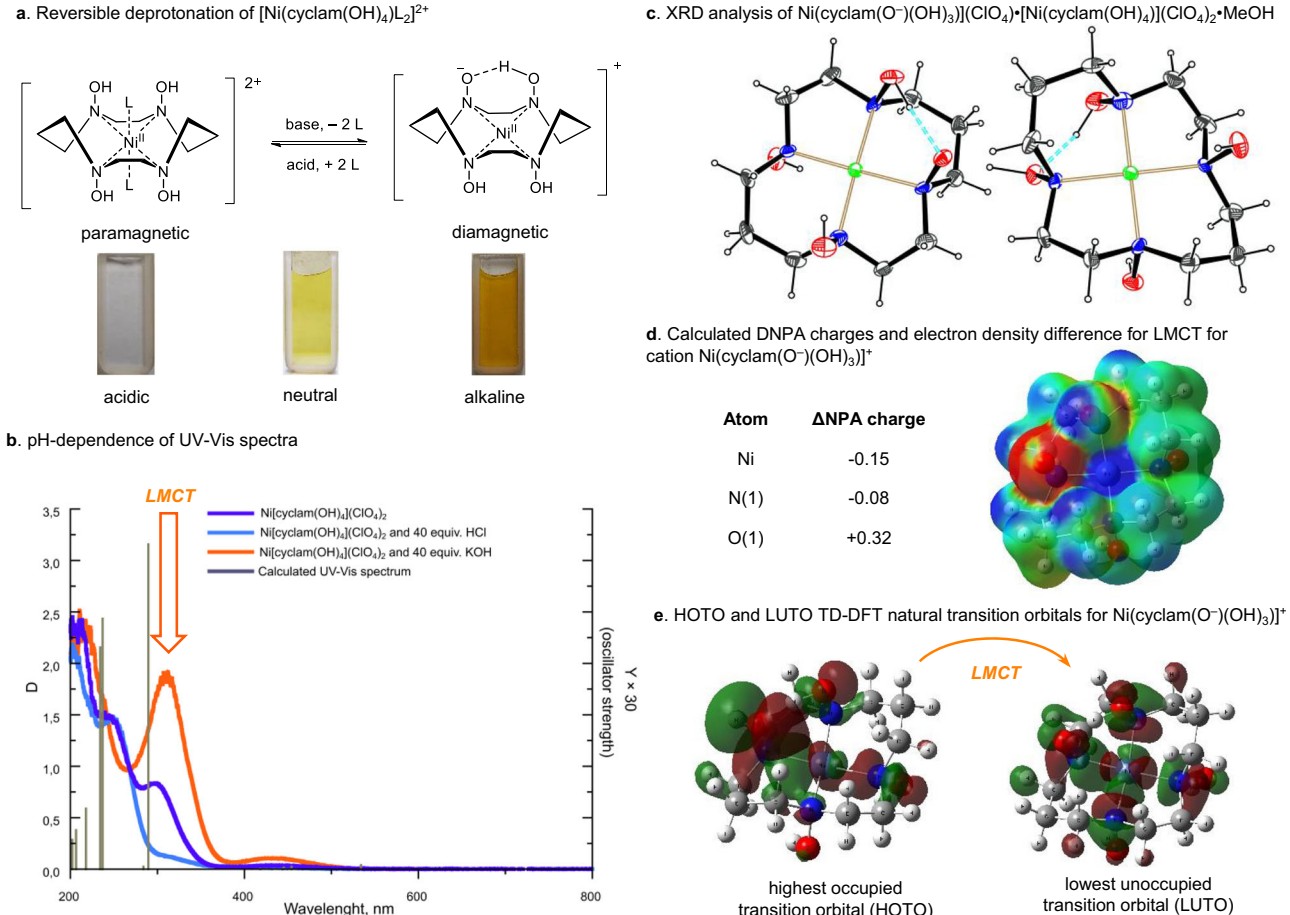

**a.** Reversible deprotonation of [Ni(cyclam(OH)₄)L₂]²⁺

paramagnetic                    diamagnetic

acidic            neutral            alkaline

**b.** pH-dependence of UV-Vis spectra

LMCT

Ni[cyclam(OH)₄](ClO₄)₂
Ni[cyclam(OH)₄](ClO₄)₂ and 40 equiv. HCl
Ni[cyclam(OH)₄](ClO₄)₂ and 40 equiv. KOH
Calculated UV-Vis spectrum

**c.** XRD analysis of Ni(cyclam(O⁻)(OH)₃)](ClO₄)•[Ni(cyclam(OH)₄)](ClO₄)₂•MeOH

**d.** Calculated DNPA charges and electron density difference for LMCT for cation Ni(cyclam(O⁻)(OH)₃)]⁺

| Atom | ΔNPA charge |
|------|-------------|
| Ni   | -0.15       |
| N(1) | -0.08       |
| O(1) | +0.32       |

**e.** HOTO and LUTO TD-DFT natural transition orbitals for Ni(cyclam(O⁻)(OH)₃)]⁺

LMCT

highest occupied transition orbital (HOTO)

lowest unoccupied transition orbital (LUTO)

**Fig. 9 | Studies of the pH-dependent behavior of the nickel(II)-cyclam(OH)₄ complex. a** Reversible deprotonation of Ni(cyclam(OH)₄)(ClO₄)₂ in aqueous medium. **b** Changes in UV-Vis spectra of Ni(cyclam(OH)₄)(ClO₄)₂ in the presence of a base and an acid (1.0−5.0 mM of the complex in H₂O). **c** ORTEP view of cations in [Ni(cyclam(O⁻)(OH)₃)](ClO₄)•[Ni(cyclam(OH)₄)](ClO₄)₂•MeOH (displacement ellipsoids are drawn at the 50% probability level). **d** Calculated ΔNPA charges and

isosurface of ground state electron density mapped using the value of the difference density (CI−SCF) for [Ni(cyclam(O⁻)(OH)₃)]⁺. Isovalue = 0.01. Blue region − electron density in the fifth excited state is larger than in the ground state, red region − it is smaller. **e** HOTO and LUTO TD-DFT natural transition orbitals for Ni(cyclam(O⁻)(OH)₃)]⁺.

of [Ni(cyclam(O⁻)(OH)₃)]⁺ species, in which the metal ion is more strongly bound than in the neutral complex, may explain the preferential complexation of Ni(II) with cyclam(OH)₄ over cyclam in the competition binding experiments. To the best of our knowledge, this is the first structurally characterized nickel complex with a deprotonated hydroxylamine ligand, in which the metal ion is coordinated only by nitrogen atoms.

To interpret changes in the UV-Vis spectra upon deprotonation of nickel(II)-cyclam(OH)₄ complex, quantum chemical calculation were performed. The structure of cation [Ni(cyclam(O⁻)(OH)₃)]⁺ was optimized with the TPSSh DFT functional using the ZORA scalar relativistic method and the UV-Vis absorption spectrum (vertical approximation) of [Ni(cyclam(O⁻)(OH)₃)]⁺ was calculated using QD-SC-NEVPT2 SA-CASSCF (10,9) for d-d transitions and TD-DFT with ωB97X-D for other transitions[63]. The most intense calculated d-d transition corresponds to a single electron excitation from $d_{xy}$ to $d_{x^2-y^2}$ orbital, which is typical of square-planar complexes[64]. The calculated absorption maximum occurs at 455 nm and is in good agreement with the experiment (434 nm). A characteristic feature of [Ni(cyclam(O⁻)(OH)₃)]⁺ complex is a strong absorption maximum at 309 nm (Fig. 9b) that is not observed in non-deprotonated Ni(cyclam(OH)₄)L₂ complex as well as in nickel(II)-cyclam complexes. This maximum was attributed to only one intense transition in this region (290 nm, from the ground state to the fifth excited state) in the

calculated TD-DFT UV-Vis spectrum. The calculated NPA atomic charges reveal a significant decrease in the negative charge on deprotonated oxygen (+0.32) and a reduced positive charge on Ni (−0.15). This fact indicates a charge transfer character of this transition that is also confirmed by the electron density difference between the fifth excited state and the ground state (Fig. 9d) as well as by the natural transition orbitals (Fig. 9e). Thus, the experimentally observed maximum at 309 nm was characterized as the ligand-to-metal charge transfer (LMCT) involving the oxygen atom of the deprotonated hydroxylamine group.

The facile deprotonation of the hydroxylamine unit may be a common feature of macrocyclic hydroxylamine complexes. For instance, the formation of [M(cyclam(O⁻)(OH)₃)]⁺ ions was observed in the ESI-HRMS for all the metal-cyclam(OH)₄ complexes (M = Ni, Cu, Mn and Zn). For the copper(II) complex, dianionic species were also detected in the mass-spectra. Moreover, the appearance of the aforementioned characteristic LMCT band upon addition of a base (NaHCO₃) was observed for Cu(cyclam(OH)₄)Cl₂ (see Supplementary methods). We believe that deprotonated metal-cyclam(OH)₄ complexes may have some interesting chemical properties probably resembling those of the well-known cobaloxime and nickeloxime[65,66].

### Aerobic catalytic activity of cyclam(OH)₄ complexes
Complexes of macrocyclic amines with redox-active d-metals are known to catalyze various aerobic oxidation reactions via dioxygen

activation[22–24]. Given the redox active character of hydroxylamine groups, it can be expected that the N–OH unit in crown-hydroxylamine complexes can intramolecularly assist the oxidation increasing the catalytic activity. With this in mind, we performed preliminary studies to compare the catalytic activity of cyclam and cyclam(OH)$_4$ copper complexes in aerobic oxidation. Two model catalytic reactions were approached, namely the oxidation of $p$-thiocresol to $p,p'$-ditolyl disulfide (a model of industrial Merox process[67]) and the oxidative homo-coupling of $N'$-phenylpropionohydrazide (NPPH) to give $N',N'$-diphenylpropionohydrazide (DPPH) (Fig. 10)[68]. Experiments were conducted with 5 mol% of the catalysts (Cu(cyclam(OH)$_4$)Cl$_2$ or Cu(cyclam)Cl$_2$) in the presence of a base (NaHCO$_3$) using air as the sole oxidant (aerobic atmosphere).

In the oxidation of $p$-thiocresol, both Cu(cyclam(OH)$_4$)Cl$_2$ and Cu(cyclam)Cl$_2$ proved to be active and gave a nearly quantitative yield of $p,p'$-ditolyl sulfide after 24 h. As follows from the kinetics plot shown in Fig. 10a, Cu(cyclam(OH)$_4$)Cl$_2$ was ca. three times more active than the corresponding cyclam complex. A more considerable difference was observed in the aerobic oxidation of phenyl hydrazide NPPH (Fig. 10b). Here, Cu(cyclam)Cl$_2$ showed poor activity (less than 25% yield of product and ca. 30% conversion), while Cu(cyclam(OH)$_4$)Cl$_2$ led to a 90% yield of diphenyl hydrazide DPPH after 24 h. These preliminary studies show that the Cu(II)-cyclam(OH)$_4$ complex exhibits substantially higher catalytic activity in aerobic oxidation reactions than the parent Cu(II)-cyclam complex.

To obtain mechanistic insight, control experiments and UV-Vis monitoring were performed. In the absence of a base, no conversion was detected indicating that deprotonated species may play an essential role in this process. Surprisingly, the deprotonated Cu(II)−cyclam(OH)$_4$ complex reacts with air resulting in the disappearance of characteristic features at 467 and 545 nm (Fig. 10c). The process is relatively slow and notable changes in UV-Vis are feasible only after ca. 4 h of exposure to air. Since Cu(cyclam)Cl$_2$ is unreactive towards O$_2$, it is likely that the dioxygen reactivity of deprotonated Cu(cyclam(OH)$_4$)Cl$_2$ is attributed to the irreversible oxidation of the N−O$^-$ groups via nitroxyl radicals. After 24 h under air, the complex completely disappears forming unidentified light brown-colored species that are catalytically inactive in the aerobic oxidation of NPPH (Supplementary Table 4). Based on literature analogies, we propose that NPPH reversibly binds to the Cu(II)−cyclam(OH)$_4$ complex followed by oxidation of deprotonated hydroxylamine group and intramolecular HAT between hydrazine N−H and the nitroxide N−O$^•$ units via a six-membered transition state (Fig. 10f)[30]. Thus, the NOH groups are key to turnover in the aerobic oxidation of NPPH. The azo compound PDAP (1-(phenyldiazenyl)propan-1-one) resulting from the oxidation of NPPH underwent a Cu(I)-catalyzed homo-coupling reaction with extrusion of N$_2$ and methyl propionate via a known mechanism[68].

The mechanism of the catalytic aerobic oxidation of $p$-thiocresol is different. In a 1:1 reaction of deprotonated Cu(cyclam(OH)$_4$)Cl$_2$ and $p$-thiocresol the disulfide is formed instantaneously (Fig. 10d). We believe that a common mechanism of Cu(II)-catalyzed oxidation of thiols via Cu(II)−S bond formation and its heterolytic fragmentation operates both with cyclam and cyclam(OH)$_4$ complexes (Fig. 10e)[69]. If this is true, the enhanced catalytic activity of Cu(cyclam(OH)$_4$)Cl$_2$ can be explained by a faster reoxidation of Cu(I) to Cu(II) species with air which is the rate-limiting stage of the reaction. It is likely that distinct electronic and conformational effects of cyclam(OH)$_4$ in comparison to cyclam result in a decrease of the oxidation potential of the corresponding Cu(I) complex[70].

## Discussion

In conclusion, fully $N$-hydroxylated macrocyclic amines (crown-hydroxylamines) were synthesized and studied as $N,O$-ligands in transition metal complexes. The developed straightforward approach to crown-hydroxylamines relies on a selective $N$-benzoyloxylation of parent macrocyclic polyamines (tacn, tacd, cyclam, etc.). The resulting $N$-benzoyloxylated products can serve as stable surrogates of crown-hydroxylamines due to the ease of solvolysis of the benzoate groups upon complexation of the ligand with d-metal ions. Alternatively, deprotection can be accomplished by hydrozinolysis to give free hydroxylamines. The structure and chemical properties of two model macrocyclic hydroxylamines, tacn(OH)$_3$ and cyclam(OH)$_4$, were explored. Key features that differ macrocyclic hydroxylamines from parent polyamines were identified, namely:

1. Macrocyclic hydroxylamines form multiple strong intramolecular hydrogen bonds that contribute to the stability of free ligands, their salts, and the corresponding metal complexes. These hydrogen bonds govern the formation of unusual conformers, tautomers (the $N$-oxide form of the hydroxylamine group), and H-bonded supramolecular structures.

2. Unlike macrocyclic amines (in particular, cyclam), crown-hydroxylamines are less prone to stabilize higher oxidation states of d-metals. Due to this, air-stable complexes of Mn(II) with cyclam(OH)$_4$ can be prepared, which are unstable for cyclam.

3. Transition metal complexes of crown-hydroxylamines (cyclam(OH)$_4$) easily undergo deprotonation of the hydroxylamine unit to form intensely colored species, such as [M(cyclam(O$^-$)(OH)$_3$)]$^+$. The structure of these species resembles metal dioximates (e.g., cobaloxime and nickeloxime) having a strong intramolecular O−H...O$^-$ bonding. In the deprotonated complexes, the metal ion is more strongly bound than in the neutral complex and in the corresponding cyclam complex.

4. Transition metal complexes of crown-hydroxylamines exhibit catalytic activity in dioxygen activation. For instance, the copper(II)-cyclam(OH)$_4$ complex shows promising catalytic activity in the aerobic oxidation of S−H and N−H bonds, while the corresponding cyclam complex shows poor performance in these processes. Since the presence of a base is essential for the catalytic activity, deprotontated copper-cyclam(OH)$_4$ species are believed to play the key role in the dioxygen activation.

Overall, the unusual structural and reactivity features of macrocyclic hydroxylamines can be attributed to the formation of extensive hydrogen-bond networks and deprotonation of the hydroxylamine groups that enhances the electron-donating ability of the ligand. We believe that these properties may have useful applications, in particular in catalysis and in metal ion sensing. Thus, crown-hydroxylamines could become a good supplement to the variety of macrocyclic amine ligands available in the chemist's toolbox.

## Methods

For general experimental, instrumental, and computational methods, detailed synthetic procedures, and full compound characterization, see Supplementary methods.

### Materials

All the reagents used in this work were purchased from commercial sources and were used as received. The synthesis of commercially unavailable starting macrocyclic polyamines was performed according to literature protocols with slight modifications (see Supplementary methods).

### General procedure for the synthesis of $N$-benzoyloxylated macrocyclic polyamines (method A)

Into a round-bottom flask were placed a solution (25 mM) of a cyclic polyamine (1 equiv.) in CH$_2$Cl$_2$ and Cs$_2$CO$_3$ (3×n equiv., n − number of nitrogen atoms in the starting polyamine). To the resulting stirred suspension was added (PhCOO)$_2$ (2×n equiv.) followed by water (9×n equiv.). The reaction mixture was stirred at room temperature

**a.** Catalytic aerobic oxidation of *p*-thiocresol

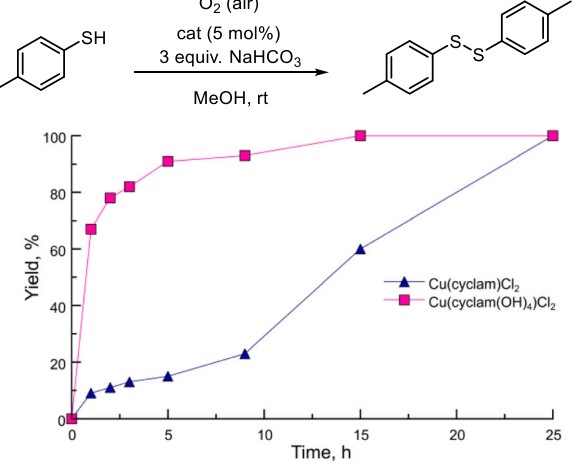

**b.** Catalytic aerobic homo-coupling of *N'*-phenylpropionohydrazide

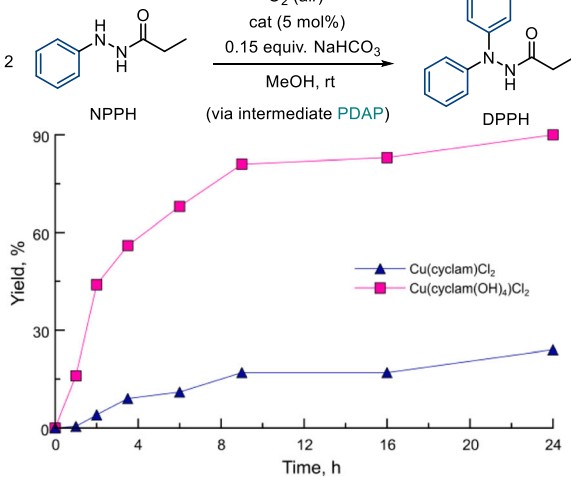

**c.** Reaction of deprotonated Cu(cyclam(OH)₄)Cl₂ with air (UV-Vis)

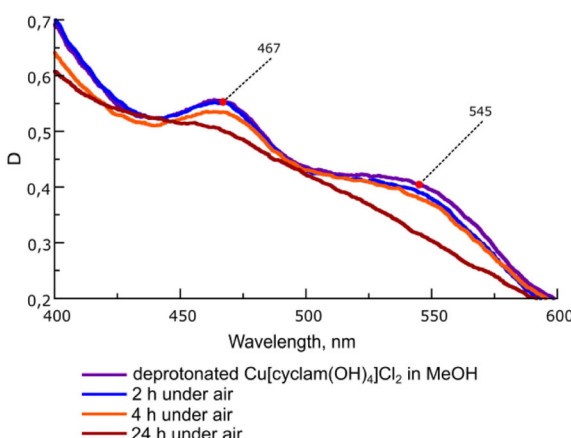

**d.** Reaction of deprotonated Cu(cyclam(OH)₄)Cl₂ with *p*-thiocresol

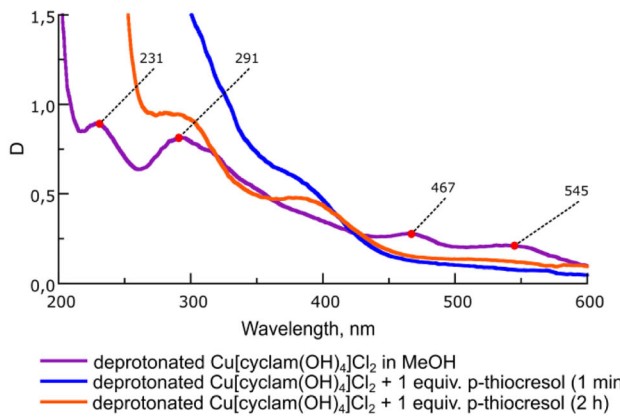

**e.** Proposed mechanism for oxidation of *p*-thiocresol

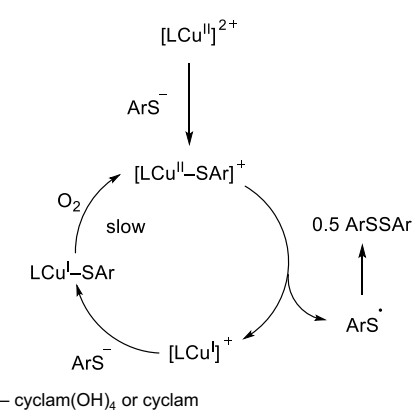

**f.** Proposed mechanism for oxidation of *N'*-phenylpropionohydrazide

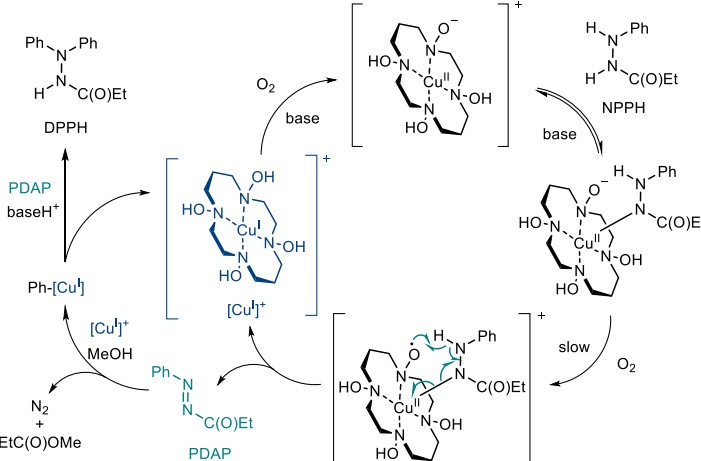

**Fig. 10 | Catalytic activity of Cu(cyclam(OH)₄)Cl₂ and Cu(cyclam)Cl₂ in aerobic oxidation reactions. a** Catalytic aerobic oxidation of *p*-thiocresol. **b** Catalytic aerobic homo-coupling of *N'*-phenylpropionohydrazide. **c** Reaction of deprotonated Cu(cyclam(OH)₄)Cl₂ with air (UV-Vis). **d** Reaction of deprotonated Cu(cyclam(OH)₄)Cl₂ with *p*-thiocresol (UV-Vis). **e** Proposed mechanism for oxidation of *p*-thiocresol. **f** Proposed mechanism for oxidation of *N'*-phenylpropionohydrazide.

overnight. After reaction completion (TLC control), water (amount equal to the reaction mixture volume) was added. The crude product was extracted with CH₂Cl₂. The organic phase was washed with saturated NaCl solution, dried (Na₂SO₄) and evaporated under reduced pressure. The product was purified by column chromatography.

**General procedure for the synthesis of *N*-benzoyloxylated macrocyclic polyamines (method B)**

Into a round-bottom flask were placed a solution (25 mM) of a cyclic polyamine (1 equiv.) in CH₂Cl₂, Cs₂CO₃ (1 equiv.) and finely ground K₂CO₃ ([3 × n−1], n − number of nitrogen atoms in the starting

polyamine). To the resulting stirred suspension was added $(PhCOO)_2$ ($2 \times n$ equiv.) followed by water ($20 \times n$ equiv., final concentration of $M_2CO_3$ – ca. 8 M). The reaction mixture was stirred at room temperature overnight. After reaction completion (TLC control), water (amount equal to the reaction mixture volume) was added. The crude product was extracted with $CH_2Cl_2$. The organic phase was washed with saturated NaCl solution, dried ($Na_2SO_4$) and evaporated under reduced pressure. The product was purified by column chromatography.

### 1,4,8,11-Tetraazacyclotetradecane-1,4,8,11-tetraol (cyclam(OH)$_4$)

To a stirred solution of 1,4,8,11-tetrabenzoyloxy-1,4,8,11-tetraazacyclotetradecane cyclam(OBz)$_4$ (670 mg, 0.98 mmol, 1 equiv.) in $CHCl_3$ (17 ml) was added hydrazine hydrate (1.0 g, 20 mmol, 20 equiv.) (Caution: Hydrazine is both highly toxic and reactive and must be handled using appropriate protective equipment to prevent physical contact with either vapor or liquid.). The reaction mixture was refluxed until full consumption of the starting material (4 h, TLC control) and formation of precipitate. The mixture was cooled to rt, and the volatiles were removed under reduced pressure. The crude product was well dried from water and residual hydrazine at ca. 0.5 Torr. The residue was suspended in $CHCl_3$ (5 ml), filtered, washed with $CHCl_3$ ($3 \times 2$ ml), and dried (0.5 Torr) until constant weight. Yield: 150 mg (58%). White solid. Mp = 210–213 °C. $^1$H NMR (300 MHz, $D_2O$) δ, ppm: 1.62–1.84 (m, 2H, $NCH_2CH_2CH_2N$), 2.44–2.69 (m, 2H, $NCH_2CH_2CH_2N$), 2.80–2.97 (m, 4H, $NCH_2CH_2CH_2N$), 3.13–3.36 (m, 8H, $NCH_2CH_2N$), 3.39–3.62 (m, 4H, $NCH_2CH_2CH_2N$). $^{13}$C NMR (75 MHz, $D_2O$) δ, ppm: 19.8 ($CH_2$), 55.9 ($CH_2CH_2CH_2N$), 61.0 ($CH_2N$). FT-IR (KBr): 3450 (s, sh), 3152 (m, br), 2975 (s), 2929 (m, sh), 2849 (s, br), 1640 (w, br), 1450 (s, sh), 1353 (s, sh), 1310 (m), 1278 (m), 1248 (w, sh), 1159 (s), 1091 (s, sh), 985 (m), 943 (s), 913 (s), 875 (m), 843 (s), 758 (w), 720 (s), 616 (m), 566 (m), 533 (m), 484 (s, sh), 434 (s, sh). ESI-HRMS m/z: [M+H]$^+$ Calcd for $[C_{10}H_{25}N_4O_4]^+$ 265.1870; Found 265.1866. Anal. Calcd for $C_{10}H_{24}N_4O_4$: C, 45.44; H, 9.15; N, 21.20. Found: C, 45.28; H, 8.98; N, 21.22.

### General procedure for the synthesis of cyclam(OH)$_4$ complexes

To a solution of cyclam(OH)$_4$ (20 mg, 76 µmol, 1 equiv.) in boiling *i*-PrOH (ca. 4 ml) was added a solution of the corresponding d-metal salt MX$_2$ in *i*-PrOH (1 ml). The resulting hot solution was transferred into a Petri dish and the solvent was slowly evaporated under gentle heating (45 °C) until crystals were formed. Then, heating plate was removed and the rest of the solvent was allowed to evaporate at ambient temperature. The residual solid was washed twice with cooled (ca. 15 °C) *i*-PrOH (1.5 ml) and centrifuged. The resulting crystalline complex was dried under reduced pressure (c.a. 0.5 Torr) until constant weight. For most of the complexes, crystals for X-ray diffraction analysis were obtained by slow vapor diffusion of diethyl ether into a methanolic solution of the complex.

## Data availability

The X-ray crystallographic coordinates for the structures reported in this article have been deposited at the Cambridge Crystallographic Data Centre (CCDC) under deposition numbers: CCDC 2265473 (tacn(OBz)$_3$), CCDC 2257251 (cyclam(OBz)$_4$), CCDC 2265480 (cyclam(OH)$_4$·HCl·MeOH), CCDC 2257250 (cyclam(OH)$_4$·2HBr), CCDC 2257253 ([Ni$_2$(µ-Cl)(µ-O$_2$CPh)(tacn(OH)$_3$)$_2$Cl$_2$]), CCDC 2257254 ([Zn(tacn(OH)$_3$)$_2$](NO$_3$)$_2$), CCDC 2257248 (Cu(cyclam(OH)$_4$)Cl$_2$), CCDC 2257249 (Mn(cyclam(OH)$_4$)Br$_2$·1.33 cyclam(OH)$_4$), CCDC 2257252 (Zn(cyclam(OH)$_4$)Cl$_2$), CCDC 2265474 (Mn(cyclam(OH)$_4$)Cl$_2$), CCDC 2265481 (Ni(cyclam(OH)$_4$)(NO$_3$)$_2$), CCDC 2259686 ([Ni$_2$(µ-Cl)$_2$(tacn)$_2$Cl$_2$]), CCDC 2259687 ([Ni(tacn)$_2$]Cl$_2$), and CCDC 2271901 (Ni(cyclam(OH)$_4$)(ClO$_4$)$_2$·Ni(cyclam(OH)$_3$(O$^-$))(ClO$_4$)·MeOH). The data can be obtained free of charge from The Cambridge Crystallographic Data Centre [http://www.ccdc.cam.ac.uk/data_request/cif]. The data

generated in this study (copies of NMR, UV-Vis, FT-IR spectra, HRMS, raw data for kinetic traces, cyclic voltammograms, X-ray crystallographic data and refinement details, data generated from DFT calculation) are provided in the Supplementary Information file. Source data for Figs. 8–10 and Cartesian atomic coordinates for optimized structures are provided with this paper. All other data are available from the corresponding author upon request. Source data are provided with this paper.

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

## Acknowledgements

The authors are grateful to the Department of Structural Studies of Zelinsky Institute of Organic Chemistry for performing the HRMS measurements (Dr. Natalya Kolotyrkina and Dr. Alexander Chizhov) and X-ray diffraction studies for compounds tacn(OBz)$_3$, cyclam(OH)$_4$•HCl•MeOH, Mn(cyclam(OH)$_4$)Cl$_2$, Ni(cyclam(OH)$_4$)(NO$_3$)$_2$ (Dr. Mikhail Minyaev, Mrs. Darina Nasyrova, and Mr. Roman Dolotov). X-ray diffraction data for other compounds were collected using the equipment of Center for molecular composition studies of INEOS RAS.

## Author contributions

V.K.L. performed all the experimental work, recorded and analyzed NMR, cyclic voltammograms (CV), UV-Vis, and FT-IR spectra; I.S.G. performed DFT, TD-DFT and CASSCF calculations; Y.V.N. and S.A.A. performed single crystal X-ray diffraction studies; A.Y.S. conceived and designed the work, supervised the project and wrote the manuscript with input of all co-authors.

## Competing interests

The authors declare no competing interests.
