## [Peer Review File · Nature Communications]

Crown-hydroxylamines are pH-dependent chelating N,O-ligands with a potential for aerobic oxidation catalysisReviewers' Comments:

Reviewer #1:

Remarks to the Author:

The authors report the synthesis of a large, new family of macrocyclic ligands bearing N-OH moieties. The overall work is thorough, particularly the novel coordination chemistry and crystallographically characterized structures, and there is a lot of potential for some really novel chemistry. The report, although enjoyable to read and see the large amount of synthetic work required to bring this ligand class to fruition, doesn't really report as much fundamentally new or surprising chemistry. There are some catalytic reactions reported using the crown-hydroxylamine ligands, but it is rather brief and represents a small portion of the manuscript. Again, I really like the science, but as for impact or significance of the work presented in *this* manuscript, the case isn't quite made that these are game-changing to inorganic chemistry or catalysis. It is, of course, totally possible that there is really important science to be found in this area, but what was reported is essentially just the synthesis, conformational complexity, and coordination chemistry of the ligands and a preliminary foray into catalytic oxidation.

The presence of the N-OH functionalities is extremely intriguing for oxidation chemistry, particularly if the N-O bond can intramolecularly assist with oxidation, so it's a little surprising to see that this wasn't really mentioned. I'm guessing that that is part of a more in-depth study, particularly as it pertains to the irreversible oxidation behavior in cyclic voltammetry experiments. The catalytic data were also sparse but an intriguing start - to what extent do the N-OH groups participate in mediating reactions with O₂? Hydroxylamine protons are notoriously easy to abstract homolytically, leading to potentially very interesting kinetics with O₂ in solution (as well as any other radical intermediates), but none of that was explored in the present manuscript.

On the subject of the electrochemistry, I was a little surprised at how the CV results were presented - normally, at least in my experience, full CVs are taken at the start just to get a feel for where the redox events are, and then the redox events are isolated and studied individually as a function of scan rate (and pH, ions, etc). For example, in Figure 8a/S15B, the scan begins at -0.5V in the anodic direction and begins to show evidence of an oxidation event at +0.25V; then, during the next cycle, what looks to be the same (or similar) event appears again, but cathodically shifted to -0.1V (and all subsequent cycles show the event there). Something like that could be due to a lot of things, such as the working electrode surface changing or the reference electrode being initially unstable, and one would work through the possibilities to figure out what's happening. In isolation, irreversible waves are very difficult to interpret due to the conflation of kinetic and thermodynamic parameters (e.g. an EC or ECE mechanism leading to cathodic/anodic shifting), so I'd caution against comparing irreversible events from one complex to irreversible events of another complex (e.g. Mn(cyclam) and Mn(cyclam-OH)).

Reviewer #2:

Remarks to the Author:

In this work, the authors have prepared a dozen of aza-macrocycles in which all amine functions were turned into hydroxylamines (the "crown-hydroxylamines" in the title), prepared a handful of metal complexes with these coordinating macrocycles and studied the physico-chemical properties of some of the ligands and complexes. The experiments are well performed and well presented.

The synthetic methodology to install the hydroxylamine functions is known (ref 49-50), but applying it to polyamines required optimization to ensure that every function is hydroxylated. Getting decent yields is therefore a tour de force by the authors, in addition to being the first time such poly-hydroxylamine macrocycles (about a dozen in total) are made.

The structural and acid-base properties of the metal complexes are well described and interesting since there are very few complexes of this kind. Catalytic behaviour is reported, but only for two test reactions and without application as a useful synthetic methodology.

Overall, this is a very fine paper, but it remains primarily an "inorganic chemistry" paper with few tangents to organic chemistry, catalytic methodology, materials, let alone other fields of science or addressing social issues (e.g. the UN SDGs). Thus, I am not sure if Nature Communications is the best venue for this work. In addition, there are a few issues that need to be fine-tuned.

1. In the introduction, you need to make a difference between hydroxylamines and hydroxamic acids. They are very different species. The difference is even more pronounced than when comparing amines and amides. For example, hydroxamic acids (once deprotonated) are excellent ligands for iron, but (primary) hydroxylamines decompose readily in the presence of iron (Fenton-like behaviour, disproportionation...).

2. The electrochemical experiments do display electron transfer events, but there is no characterization of the products of these electron transfers. For example in Figure 8, are the reduction waves (around -0.66 and -1.16 V) the cathodic component ("return") from the oxidation wave or are these waves already present when the potential is scanned, from the resting potential, in the direction of reductions (towards negative potentials)? No electrolysis or spectro-electrolysis was performed and, therefore, there is no evidence for the true nature of the redox partners. The E/C nature of the events (electron transfer coupled, or not, with a chemical reaction) was not discussed either.

3. In Figure 10, the catalytic traces are based on three points only. For a 24-hour reaction, this is greatly insufficient. A kinetic trace cannot be a mere trend line.

Small typo: page 4, first paragraph of "Results", line 5: Cyclam is a tetramine, not a "triamine" (as written).

Bottomline: these crown-hydroxylamines are very new, but the authors could not demonstrate a truly impactful application that would warrant publication at the very top. Even with the novelty of these species, the interesting acid-base properties of the complexes, the precision in the work done, I fear that the readership will remain limited to coordination chemists.

Reviewer #3:

Remarks to the Author:

I enjoyed reading the results reported in this manuscript. The authors have done a good job revealing the interesting reactivities and structures made possible with the new ligands they have introduced, and I believe that this family of ligands and the complexes they will form stand good potential for promoting/enabling exciting new structures, material properties and catalytic reactivities in a number of areas such as coordination chemistry, photochemistry, catalysis, redox catalysis, etc. This said, I have some reservations about whether the report is a good fit for Nature Chem. My arguments are outlined below.

To begin with, the ligands reported here are new, but one could argue that they are not quite unprecedented as they can be considered to represent variations on the ligand (tfoH3) reported by the authors in 2021. This is not to deny that the presence of multiple OH motifs in a large macrocycle might have greater impact than in (tfoH3), but rather to emphasize that calling the new ligands "unprecedented" demands a higher degree of veracity which is not met in my view.

Similarly, the tautomerism of hydroxylamine to aminoxide coordination observed in the new ligands might well be quite different from previously reported cases of such behavior, but it should be recognized that such tautomerism is not unknown and so once again describing this as "unprecedented" for the present case seems somewhat premature and inappropriate.

Finally, the catalytic oxidations enabled by the new Cu/cyclam(OH)₄ complexes don't strike this reader as game-changing or exciting new applications or even significantly improved reactivity over existing technology, though I agree that it is encouraging to see that the analogous Cu/cyclam precursors are much less active in the promotion of these reactions. For reference, if we compare these results to a few cases of significant new levels of catalytic reactivities reported over the past 2-3 decades in journals such as Science and Nature, I would say that one needs orders of magnitude improvements in reactivity or selectivity to make a strong case for publication in very selective venues.

Assuming the above are fair assessments of the significance of the results disclosed here, the question arises if the combined progress represented in the three aspects mentioned above, i.e., "newish" ligands and coordination behavior, and good but not huge degrees of improved catalytic activities, warrants publication in Nature Chem. This is a tough question to answer for someone like me who is not a regular reader of this journal, and in this sense, I defer the answer to the editor(s). It might well be that the novelty and significance of the results need not be at the same level as one would expect for a journal such as Nature, and so my hesitation might be out of place here. Nevertheless, it seems to me that the results presented here would not be out of place in one of the top general chemistry journals such as JACS, Angew, Chemical Science, etc.

Aside from the above reflections on the significance of the results reported, I wish to add that the experimental work has been conducted with expertise, the results have been interpreted logically and the conclusions are valid. Moreover, the manuscript has been prepared with high degree of professionalism and attention to details, the bibliography is very good, and I didn't find any major substantive or stylistic issue to require correction. A couple of minor revisions could be to shorten the Introduction section (which is too long for this type communication). Also, on the second last line of Intro, I wonder if the word "albeit" should be replaced by "whereas". And two final questions for the authors:

(1) Was the reactivity of tacn(OH)₃ tested with metals such as Cu and Mn, or did you really limit your exploration to only Zn and Ni? It would be curious to see how differently the larger macrocycles react with these metals.

(2) Did you explore the photochemistry of the Ni(cyclam(OH)₄) complexes at low temperatures? I am particularly curious if any luminescence might be observable.

Point-to-Point Response to Reviewers' Comments

Response to Reviewer #1

Reviewer's comment #1: The presence of the N-OH functionalities is extremely intriguing for oxidation chemistry, particularly if the N-O bond can intramolecularly assist with oxidation, so it's a little surprising to see that this wasn't really mentioned. I'm guessing that that is part of a more in-depth study, particularly as it pertains to the irreversible oxidation behavior in cyclic voltammetry experiments.

Our response: Thank you for pointing out this issue. Indeed, the intramolecular assistance of the N-OH groups in the catalytic oxidation is expected to be one of the most intriguing things about our N,O-ligands (yet it wasn't directly mentioned in the original manuscript). In the revised manuscript, the sentence "In these catalysts, the N–O bond can intramolecularly assist the oxidation of an organic substrate via nitroxyl radical species" was added together with citations of literature precedents of this phenomenon. Also, a discussion of the possible participation of N-OH groups in catalytic aerobic oxidation was included (see answer to comment #2).

Reviewer's comment #2: The catalytic data were also sparse but an intriguing start - to what extent do the N-OH groups participate in mediating reactions with O₂? Hydroxylamine protons are notoriously easy to abstract homolytically, leading to potentially very interesting kinetics with O₂ in solution (as well as any other radical intermediates), but none of that was explored in the present manuscript.

Our response: Thank you for this suggestion. Indeed, the role of hydroxylamine groups in these catalytic reactions is intriguing and this will be a subject of further in-depth studies. Nevertheless, to obtain a preliminary mechanistic insight, control experiments and UV-Vis monitoring were performed. The key observation was that the deprotonated Cu(II)–**cyclam(OH)**₄ complex slowly reacts with air (a parent Cu(II)–**cyclam** is unreactive towards O₂). Together with the revised electrochemistry studies, this indicates an irreversible oxidation of the N–O[•] groups via nitroxyl radicals. The oxidized species are catalytically inactive in the aerobic oxidation of hydrazide **NPPH**. Based on literature analogies (*J Am Chem Soc*, **2020**, *142*, 19023), we propose that the hydrazide anion reversibly binds to the Cu(II)–**cyclam(OH)**₄ complex followed by oxidation of deprotonated hydroxylamine group and intramolecular HAT between hydrazide N–H and the nitroxide N–O[•] units via a six-membered transition state (see Figure 10, f in the revised manuscript). To confirm this mechanism, a macrocyclic hydroxylamine ligand bearing alkyl groups at the alpha position is needed (to stabilize nitroxyl radicals by preventing H-abstraction leading to nitrones). Further work in this direction will be performed.

The aerobic oxidation of *p*-thiocresol was shown to be much faster than the oxidation of the Cu(II)–***cyclam*(OH)₄** complex. Therefore, we suppose that a common mechanism of Cu(II)-catalyzed oxidation of thiols via Cu(II)–S bond formation and its heterolytic fragmentation operates both with ***cyclam*** and ***cyclam*(OH)₄** complexes (see Figure 10, e in the revised manuscript). The enhanced catalytic activity of the Cu(II)–***cyclam*(OH)₄** complex can be thus explained by a faster reoxidation of Cu(I) to Cu(II) species with air which is the rate-limiting stage of the reaction. It is likely that distinct electronic and conformational effects of ***cyclam*(OH)₄** in comparison to ***cyclam*** result in a decrease of the oxidation potential of the corresponding Cu(I) complex (the conformation of macrocyclic ligand was previously shown to have a significant effect on the dioxygen reactivity of Cu(I) complexes, see *Chem Eur J*, **2016**, *22*, 5133). Thus, NOH groups may not directly participate in the Cu(II)–***cyclam*(OH)₄** catalyzed oxidation of thiols.

A discussion of these experiments and the proposed mechanisms was added in the revised manuscript. Figure 10 was revised accordingly.

Reviewer's comment #3: On the subject of the electrochemistry, I was a little surprised at how the CV results were presented - normally, at least in my experience, full CVs are taken at the start just to get a feel for where the redox events are, and then the redox events are isolated and studied individually as a function of scan rate (and pH, ions, etc). For example, in Figure 8a/S15B, the scan begins at -0.5V in the anodic direction and begins to show evidence of an oxidation event at +0.25V; then, during the next cycle, what looks to be the same (or similar) event appears again, but cathodically shifted to -0.1V (and all subsequent cycles show the event there). Something like that could be due to a lot of things, such as the working electrode surface changing or the reference electrode being initially unstable, and one would work through the possibilities to figure out what's happening. In isolation, irreversible waves are very difficult to interpret due to the conflation of kinetic and thermodynamic parameters (e.g. an EC or ECE mechanism leading to cathodic/anodic shifting), so I'd caution against comparing irreversible events from one complex to irreversible events of another complex (e.g. Mn(*cyclam*) and Mn(*cyclam*-OH)).

Our response: Thank you for pointing out this issue. CVs were retaken to isolate the redox events (observed in full CVs) and to study them individually as a function of scan rate. New CV data were included in the Supplementary material (pages S151-S162, Supplementary Figures 15-25). Also, the final products of the electrochemical oxidation of ***cyclam*(OH)₄** were proposed based on ESI-HRMS and NMR. In the manuscript, Figure 8 was revised and characteristic CVs depicting separately oxidation and reduction processes were given instead of full CVs. The discussion of cyclic voltammetry was revised to avoid the comparison of irreversible events for different complexes.

Response to Reviewer #2

Reviewer's comment #1: In the introduction, you need to make a difference between hydroxylamines and hydroxamic acids. They are very different species. The difference is even more pronounced than when comparing amines and amides. For example, hydroxamic acids (once deprotonated) are excellent ligands for iron, but (primary) hydroxylamines decompose readily in the presence of iron (Fenton-like behaviour, disproportionation...).

Our response: Thank you for pointing out this issue. We agree that hydroxylamic acids are quite different from hydroxylamines, especially in their redox behavior and coordination chemistry. Therefore, all references to hydroxamic acids and related siderophores and chelators were removed from the Introduction text and Figure 1.

Reviewer's comment #2: The electrochemical experiments do display electron transfer events, but there is no characterization of the products of these electron transfers. For example in Figure 8, are the reduction waves (around -0.66 and -1.16 V) the cathodic component ("return") from the oxidation wave or are these waves already present when the potential is scanned, from the resting potential, in the direction of reductions (towards negative potentials)? No electrolysis or spectro-electrolysis was performed and, therefore, there is no evidence for the true nature of the redox partners. The E/C nature of the events (electron transfer coupled, or not, with a chemical reaction) was not discussed either.

Our response: Thank you for pointing out this issue. CVs were retaken to isolate the redox events and to study them individually as a function of scan rate. In the manuscript Figure 8 was revised and characteristic CVs depicting separately oxidation and reduction processes were given instead of full CVs. New CV data were included in the Supplementary material (pages S151-S162, Supplementary Figures 15-25).

CV studies show that macrocyclic hydroxylamines and their complexes exhibit a complicated irreversible redox behavior. Characterization of multiple species involved in these E_rC_i processes is a tough task, and electrolysis coupled with UV-Vis spectra may not give enough information about the structure of these species. To identify the final products of these electron transfers, electrochemical oxidation of ***cyclam(OH)*₄** was performed followed by ESI-HRMS analysis. Ions corresponding to di-, tetra-, hexa- and octa-dehydrogenated ligand (L-2H, L-4H, L-6H, and L-8H) were identified. The same species were detected in a chemical oxidation of ***cyclam(OH)*₄** by AgNO₃. NMR analysis showed signals around 7.5 ppm (protons) and 130 ppm (carbon) attributed to the double C,N-bonds. From these data, it can be suggested that ***cyclam(OH)*₄** undergoes multiple irreversible oxidations of the hydroxylamine units to nitrones via nitroxyl radicals followed by abstraction of α -C-H hydrogens. However, attempts to isolate these oxidized products were unsuccessful most likely due to their instability. A discussion of these experiments was included in the manuscript text.

Reviewer's comment #3: In Figure 10, the catalytic traces are based on three points only. For a 24-hour reaction, this is greatly insufficient. A kinetic trace cannot be a mere trend line.

Our response: Thank you for pointing out this issue. These experiments were repeated and more data points were collected. Figure 10 was revised accordingly, and a mere trend line was removed. Also, source data were provided in xlsx format.

Reviewer's comment #4: Small typo: page 4, first paragraph of "Results", line 5: Cyclam is a tetramine, not a "triamine" (as written).

Our response: Thank you. This mistake was corrected.

Response to Reviewer #3

Reviewer's comment #1: To begin with, the ligands reported here are new, but one could argue that they are not quite unprecedented as they can be considered to represent variations on the ligand (tfoH₃) reported by the authors in 2021. This is not to deny that the presence of multiple OH motifs in a large macrocycle might have greater impact than in (tfoH₃), but rather to emphasize that calling the new ligands "unprecedented" demands a higher degree of veracity which is not met in my view. Similarly, the tautomerism of hydroxylamine to aminoxide coordination observed in the new ligands might well be quite different from previously reported cases of such behavior, but it should be recognized that such tautomerism is not unknown and so once again describing this as "unprecedented" for the present case seems somewhat premature and inappropriate.

Our response: Thank you for this comment. To avoid controversy over the structural novelty of these ligands and *N*-oxide tautomerism, words like "unprecedented", "novel", "new", and "for the first time" were avoided in the manuscript text.

Reviewer's comment #2: A couple of minor revisions could be to shorten the Introduction section (which is too long for this type communication).

Our response: Thank you for this suggestion. The introduction section was shortened by removing some less relevant discussions (in particular, text describing nickel and zinc complexes of cyclam, natural siderophores and related artificial chelators were removed).

Reviewer's comment #3: Also, on the second last line of Intro, I wonder if the word "albeit" should be replaced by "whereas".

Our response: Thank you. This was corrected in the manuscript text and in the abstract.

Reviewer's comment #4: Was the reactivity of tacn(OH)₃ tested with metals such as Cu and Mn, or did you really limit your exploration to only Zn and Ni? It would be curious to see how differently the larger macrocycles react with these metals.

Our response: Thank you for this suggestion. We tried to synthesize complexes of **tacn(OH)₃** with Cu(II) and Mn(II) by reacting **tacn(OBz)₃** with the corresponding metal chlorides in MeOH, yet these attempts were unsuccessful so far. In the case of copper, the removal of Bz-groups was observed by TLC, however, we failed to isolate any Cu(II)-**tacn(OH)₃** complexes from this reaction (the only crystalline material obtained was copper benzoate Cu₂(OBz)₄). With MnCl₂, no methanolysis of **tacn(OBz)₃** was detected by TLC. Thus, the synthesis of manganese-

tacn(OH)₃ complexes may require a prior generation of the free (unprotected) ligand. Further research needs to be undertaken to obtain and characterize these complexes.

Reviewer's comment #5: Did you explore the photochemistry of the Ni(cyclam(OH)₄) complexes at low temperatures? I am particularly curious if any luminescence might be observable.

Our response: Thank you for this suggestion. Room-temperature fluorescence spectra taken for a deprotonated Ni(**cyclam(OH)₄**) complex showed no emission in solution (Figure A). Unfortunately, we do not have the equipment to measure luminescence spectra at low temperatures. However, a simple test under a UV lamp ($\lambda_{\text{ex}} = 366 \text{ nm}$) does not indicate any naked-eye observable emission for the deprotonated Ni(**cyclam(OH)₄**) complex at 77 K (Figure B).

Figure. (A) Room-temperature fluorescence spectrum of Ni(**cyclam(OH)₄**)(ClO₄)₂ in $1 \cdot 10^{-5} \text{ M}$ aqueous solution at $\lambda_{\text{ex}} = 320 \text{ nm}$. (B) Photo of solid fluorescein (left) and deprotonated Ni(**cyclam(OH)₄**)(ClO₄)₂ complex (right) excited via a UV lamp ($\lambda_{\text{ex}} = 366 \text{ nm}$) at 77K.

Reviewers' Comments:

Reviewer #1:

Remarks to the Author:

The authors have largely addressed my concerns. My main issues earlier were with scope and presentation of the electrochemical studies, and the authors have responded to those. The incorporation of the proposed mechanism in Figure 10 adds a lot to the conceptual utility of the new ligand class, so that is also appreciated.

Reviewer #2:

Remarks to the Author:

Thank you for your corrections and additions. You did address some of my comments.

Overall, this is still good work, but my initial review still remains as to whether this is the appropriate journal and readership. This work deserves publication in a good journal; whether Nature Comm or elsewhere, the editor will decide.

Some further comments:

- Catalysis: did you try to fit the kinetic trace with a first-order or second-order model (linearization...)?
- The new electrochemistry experiments do add to the science, but are still quite not fully conclusive. The voltamograms are not all very clean, which renders the interpretation very difficult (provided that the samples are indeed pure).

Some minor points:

- "peraza" means **all** atoms are replaced with nitrogens. You should employ "polyaza" instead.
- Line 392 (in Figure): replace "magentic" with "magnetic"
- Line 470: The text says less than 5% yield but the figure shows about 20%. Please address.
- References 19 22 2 31 34 40 59 60 65 66 70 should list all authors, if possible. If there is space for the title of each paper, there is space for all authors to be listed.

Reviewer #3:

Remarks to the Author:

I have reviewed the revised manuscript and am satisfied that the authors have addressed my original concerns/questions.

Point-to-Point Response to Reviewers' Comments

Response to Reviewer #1

Reviewer's comment: The authors have largely addressed my concerns. My main issues earlier were with scope and presentation of the electrochemical studies, and the authors have responded to those. The incorporation of the proposed mechanism in Figure 10 adds a lot to the conceptual utility of the new ligand class, so that is also appreciated.

Our response: We are very grateful to the Reviewer for agreeing to accept our manuscript for publication in *Nature Communications*.

Response to Reviewer #2

Reviewer's comment #1: Catalysis: did you try to fit the kinetic trace with a first-order or second-order model (linearization...)?

Our response: Thank you for this suggestion. We did try to fit the obtained kinetic traces with various kinetic models. The kinetic trace of the aerobic oxidation of *p*-thiocresol with Cu-cyclam(OH)₄ catalyst best fitted the pseudo-second order kinetic model (see Figure a). However, the data obtained from the oxidation of *N*'-phenylpropionohydrazide fitted both pseudo-first and pseudo-second order models (Figure b). More in-depth kinetic studies are needed to get insights into the mechanistic details of these oxidation processes (that we are planning to perform in future).

a

b

Reviewer's comment #2: The new electrochemistry experiments do add to the science, but are still quite not fully conclusive. The voltamograms are not all very clean, which renders the interpretation very difficult (provided that the samples are indeed pure).

Our response: Thank you for pointing out this issue. We tried our best to get clear CV data for cyclam(OH)₄ and its complexes. The compounds were analytically pure, the supporting electrolyte was recrystallized several times to exclude any impurities and the solutions were thoroughly deaerated before the measurements (no odd signals were detected in CV of pure electrolyte solution). In our experiments, clear CVs were obtained for the reference and some standard compounds (ferrocene, hydroquinone, Ni(cyclam)₂Cl₂). Also, we got reproducible results for samples obtained from different experiments. We suppose that the appearance of some voltammograms might be due to a complicated redox behavior of cyclam(OH)₄, in which four hydroxylamine groups can undergo both independent and interdependent oxidation/reduction events leading to the generation of multiple species.

Reviewer's comment #3: "peraza" means *all* atoms are replaced with nitrogens. You should employ "polyaza" instead.

Our response: Thank you for pointing out this issue. We corrected term "peraza-crowns" to "polyaza-crowns" throughout the manuscript text and in Figure 1.

Reviewer's comment #4: Line 392 (in Figure): replace "magentic" with "magnetic"

Our response: Thank you. This mistake was corrected.

Reviewer's comment #5: Line 470: The text says less than 5% yield but the figure shows about 20%. Please address.

Our response: Thank you for pointing out this issue. The yield was corrected from "5%" to "25%".

Reviewer's comment #5: References 19 22 2 31 34 40 59 60 65 66 70 should list all authors, if possible. If there is space for the title of each paper, there is space for all authors to be listed.

Our response: Thank you for this suggestion. When preparing the reference list, we followed the standard *Nature* referencing style according to which all authors should be included in reference lists unless there are six or more. In the revised manuscript, we listed all authors in the references mentioned by you. However, we leave that up to the editorial office.

Response to Reviewer #3

Reviewer's comment: I have reviewed the revised manuscript and am satisfied that the authors have addressed my original concerns/questions.

Our response: We are very grateful to the Reviewer for agreeing to accept our manuscript for publication in *Nature Communications*.